# Presynaptic APP levels and synaptic homeostasis are regulated by Akt phosphorylation of huntingtin

Julie Bruyère[1†], Yah-Se Abada[2†], Hélène Vitet[1†], Gaëlle Fontaine[2], Jean-Christophe Deloulme[1], Aurélia Cès[2], Eric Denarier[1], Karin Pernet-Gallay[1], Annie Andrieux[1], Sandrine Humbert[1], Marie-Claude Potier[2], Benoît Delatour[2], Frédéric Saudou[1]*

[1]Univ. Grenoble Alpes, Inserm, U1216, CHU Grenoble Alpes, CEA, Grenoble Institut Neurosciences, Grenoble, France; [2]Institut du Cerveau et de la Moelle épinière, ICM, Inserm U1127, CNRS UMR 7225, Sorbonne Université, Paris, France

**Abstract** Studies have suggested that amyloid precursor protein (APP) regulates synaptic homeostasis, but the evidence has not been consistent. In particular, signaling pathways controlling APP transport to the synapse in axons and dendrites remain to be identified. Having previously shown that Huntingtin (HTT), the scaffolding protein involved in Huntington's disease, regulates neuritic transport of APP, we used a microfluidic corticocortical neuronal network-on-a-chip to examine APP transport and localization to the pre- and post-synaptic compartments. We found that HTT, upon phosphorylation by the Ser/Thr kinase Akt, regulates APP transport in axons but not dendrites. Expression of an unphosphorylatable HTT decreased axonal anterograde transport of APP, reduced presynaptic APP levels, and increased synaptic density. Ablating in vivo HTT phosphorylation in APPPS1 mice, which overexpress APP, reduced presynaptic APP levels, restored synapse number and improved learning and memory. The Akt-HTT pathway and axonal transport of APP thus regulate APP presynaptic levels and synapse homeostasis.

*For correspondence:
frederic.saudou@inserm.fr

†These authors contributed equally to this work

Competing interests: The authors declare that no competing interests exist.

## Introduction

Synaptic homeostasis stabilizes neural circuits and ensures faithful communication within networks that are being continuously remodeled. It involves a complex interplay between presynaptic and postsynaptic proteins that modulates synaptic morphology and strength (*Südhof, 2018*). Several studies suggest that amyloid precursor protein (APP) contributes to synapse homeostasis (for reviews see *Hoe et al., 2012*; *Müller et al., 2017*), and although the evidence is not entirely consistent, this possibility has intuitive appeal because of APP's involvement in diseases of cognition (e.g., Alzheimer's disease, Lewy body dementia, and cerebral amyloid angiopathy)(*Müller et al., 2017*). Some studies suggest that loss of APP reduces synapse density (*Weyer et al., 2014*), while others show that it increases the number of synapses (*Bittner et al., 2009*). More firmly established is the fact that APP is transported both in axons and dendrites and localizes in both the pre- and post-synaptic compartments, where it could associate with synaptic release machinery to regulate neuronal transmission (*Buggia-Prévot et al., 2014*; *Das et al., 2016*; *Fanutza et al., 2015*; *Groemer et al., 2011*; *Klevanski et al., 2015*). In addition, APP may function as an adhesion molecule at the synapse (*Müller et al., 2017*; *Soba et al., 2005*). Any modification in the transport of APP in either axons or dendrites thus has the potential to disrupt synaptic function or homeostasis. Therefore, there is a need to identify mechanisms and/or pathways that specifically regulate APP transport both in axons and/or dendrites and to determine whether manipulating these pathways control APP accumulation and synapse homeostasis.

APP is transported from the Golgi apparatus to the synapse in either dendrites or axons, and in both anterograde and retrograde directions by kinesin-1 and dynein, respectively (*Brunholz et al., 2012*; *Gibbs et al., 2015*; *Toh and Gleeson, 2016*). We and others have shown that wild-type huntingtin (HTT), but not the polyglutamine-expanded HTT that causes Huntington's disease (HD), facilitates APP transport by increasing the velocity of APP-containing vesicles (*Colin et al., 2008*; *Her and Goldstein, 2008*). HTT is a large scaffold protein that interacts with various protein complexes including molecular motor proteins and, regulates consequently the transport of several cargos (*Saudou and Humbert, 2016*). APP transport into neurites is altered upon reduction of HTT levels or by the presence of polyQ expansion on HTT (*Colin et al., 2008*; *Her and Goldstein, 2008*). However, these studies did not distinguished axons from dendrites and did not investigate the consequences on APP levels at the synapse both in vitro and in vivo. Consequently, several questions remain to be addressed regarding the interplay between HTT and APP and its physiological consequences.

To answer these questions, we studied APP and HTT in a microfluidic device that reconstitutes a corticocortical neuronal network with separate presynaptic, synaptic, and postsynaptic compartments, and further tested our findings in APPPS1 mice, which display AD-like pathology. We find that subtle modifications of axonal transport of APP change synaptic levels of APP and have dramatic consequences on synapse function.

## Results

### Developing an in vitro corticocortical network using microfluidic chambers

One of the major impediments to assessing APP transport in axons and dendrites under physiological conditions is the difficulty of recreating a mature neuronal network in a dish. Primary cultures are usually randomly distributed, with multidirectional, random connections. The use of Campenot chambers or microfluidic devices made it possible to separate axons from dendrites and soma, but neurons in these chambers are still not integrated into networks as they would be in vivo (*Taylor et al., 2005*). We therefore turned to later-generation devices (*Taylor et al., 2010*) and modified them to reconstitute an oriented network with optimized connections (*Moutaux et al., 2018*; *Virlogeux et al., 2018*). These devices contain three compartments (presynaptic, synaptic, and postsynaptic) that are fluidically isolated and separated by microchannels that are 5 µm high and 5 µm wide, but of two different lengths: 500 µm and 75 µm (*Figure 1A*). The 500 µm channels allow only axons from the presynaptic compartment to reach the synaptic compartment (*Taylor et al., 2005*). The 75 µm long microchannels allow dendrites to cross from the postsynaptic to the synaptic compartment, where MAP2 staining shows they connect with axons coming from the presynaptic compartment (*Figure 1B*). We reconstructed a corticocortical network-on-a-chip since APP protein is expressed in the cortex, and AD largely targets iso- and archicortical brain regions. In these microfluidic devices, we observed full maturation of the corticocortical neuronal network between days in vitro (DIV) 10 and 15, as revealed by uptake of FM4-64, an indicator of endocytosis/exocytosis of functional synapses (*Figure 1C*). This device is thus optimized for studying the sub-cellular dynamics of APP.

### APP is transported to synapses from both pre- and post-synaptic neurons

We transduced mouse cortical neurons at DIV eight with lentiviruses expressing APP tagged with mCherry at the C terminus (APP-mCherry), which retains APP characteristics (*Kaether et al., 2000*; *Marquer et al., 2014*). We recorded the movements of APP-mCherry vesicles at high frequency frame rate using spinning disk confocal microscopy and found that velocities reach a maximum at DIV13, when the network is fully mature, with established synapses (*Figure 1B and C*, *Figure 1—figure supplement 1*). To assess axonal transport, we first focused on the distal part of the 500 µm long microchannels (*Figure 1B*) to follow APP-mCherry vesicles within neurons transduced in the presynaptic compartment (*Figure 2A*, *Video 1*), which can be reached only by presynaptic axons (*Moutaux et al., 2018*; *Taylor et al., 2005*; *Virlogeux et al., 2018*). To assess dendritic transport of APP-mCherry vesicles, we transfected the postsynaptic neurons with a MAP2-GFP plasmid and

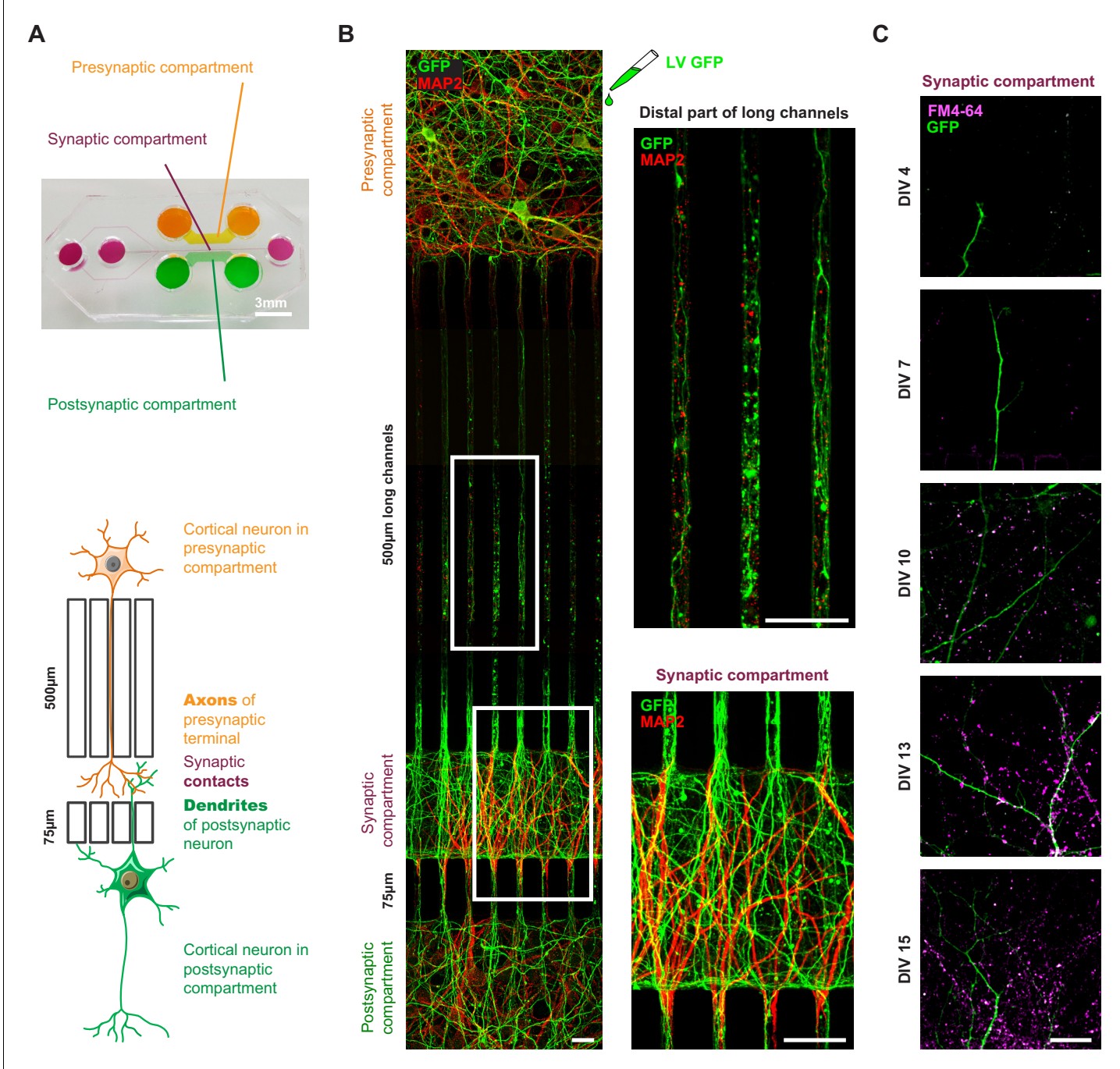

**Figure 1.** Reconstituted corticocortical mature neuronal circuit. (**A**) Image and schematic representation of the 3-compartment microfluidic chamber that allows the reconstitution of a corticocortical mature network compatible with live-cell imaging of axons and dendrites. (**B**) Presynaptic neurons were transduced with GFP (green) to visualize axons into microgrooves and MAP2 (red) immunostaining was applied on the entire microchambre at DIV13. Magnification shows axons into distal part of long microchannel but not MAP2-positive dendrites (**C**) Functional synapses were detected using FM4-64 dye (purple) that labels active presynaptic boutons on GFP dendrites (green) upon 50 mM KCl stimulation. Images represent a projection of 5 μm Z stacks. The highest number of functional corticocortical synapses is visualized between DIV10 and DIV15 in this microfluidic device. Scale bar = 20 μm. The online version of this article includes the following figure supplement(s) for figure 1:

**Figure supplement 1.** Time course of kinetics of anterograde and retrograde APP-mCherry axonal velocity after plating neurons into microchambers.

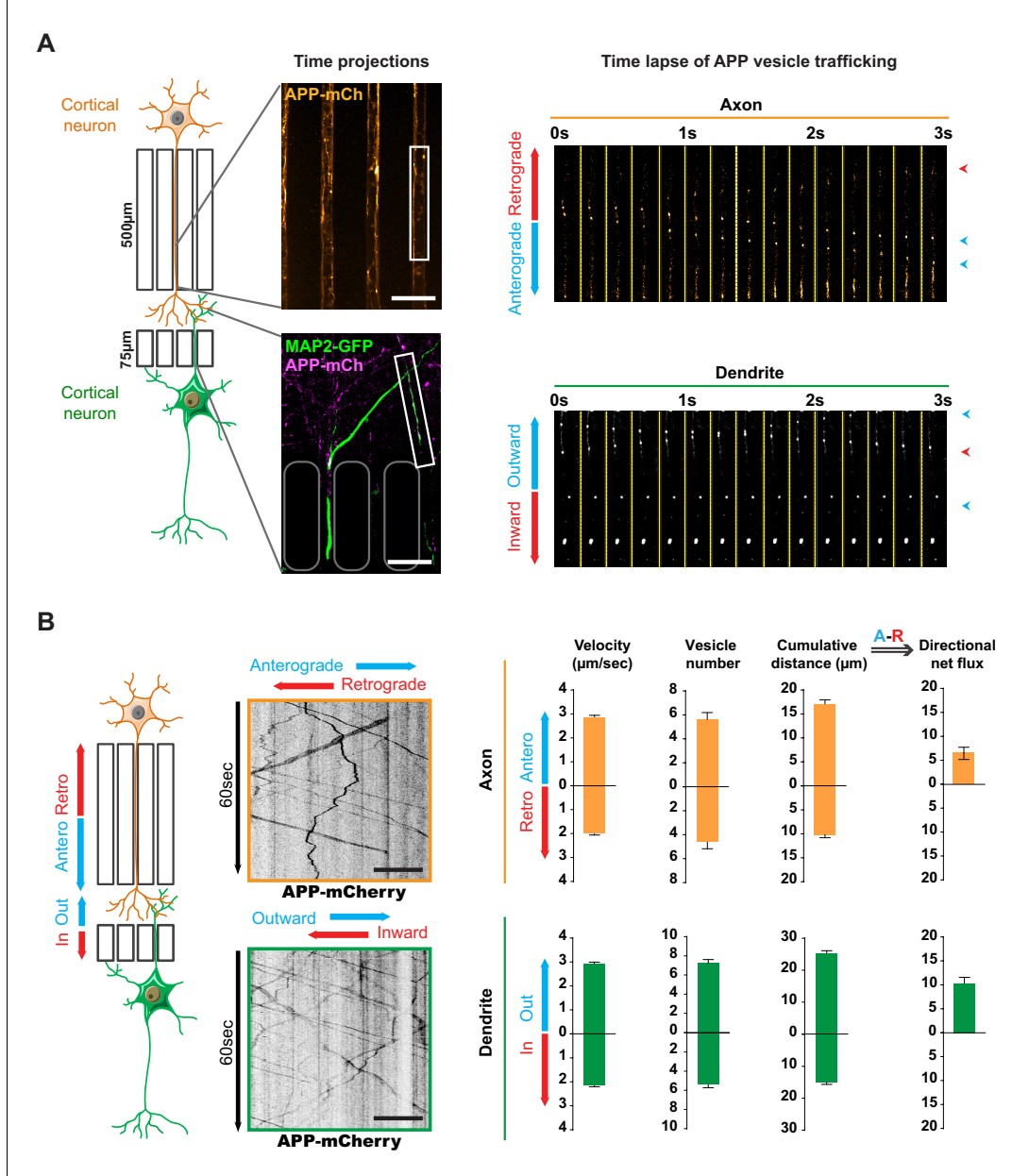

**Figure 2.** Transport of APP in axons and dendrites in reconstituted corticocortical mature neuronal circuit. (**A**) APP-mCherry was transduced into the presynaptic compartment for axonal transport analysis or into the postsynaptic compartment for dendritic trafficking. Postsynaptic neurons were transfected with MAP-2-GFP (green) to visualize dendrites that cross the short microchannels. APP-mCherry transport along the axons or the dendrites are represented in time projections of maximum signal intensities for 60 s (middle panels) and real time-lapse analysis of anterograde/retrograde or inward/outward vesicles in axons and dendrites respectively (right panel). Scale bars = 20 µm. (**B**) Kymograph analyses of APP-mCherry axonal or dendritic transport at DIV13 from time-lapse images acquired every 200 ms during 60 s. Transport characteristics such as the anterograde/retrograde or inward/outward vesicle velocities, moving vesicle number per 100 µm of neurite length, the cumulative distances travelled by vesicles and thus the directional net flux of APP-mCherry trafficking into axons (upper panel) or dendrites (lower panel) are represented by means +/- SEM of 3 independent experiments, 40 axonal and 120 dendritic axons and 674 axonal and 1160 dendritic vesicles. Scale bars = 20 µm. (see also *Videos 1* and *2*).

selected only the APP-mCherry vesicles that crossed the 75-µm-long microchannels (*Figure 2A*, *Video 2*). MAP2-GFP transfection did not modify the transport of APP by itself (data not shown).

We generated kymographs from the axonal and dendritic recordings (*Figure 2B*) and measured several transport parameters (see Materials and methods): the velocity of APP-mCherry vesicles, their number, and the cumulative distance they travelled in anterograde and retrograde directions

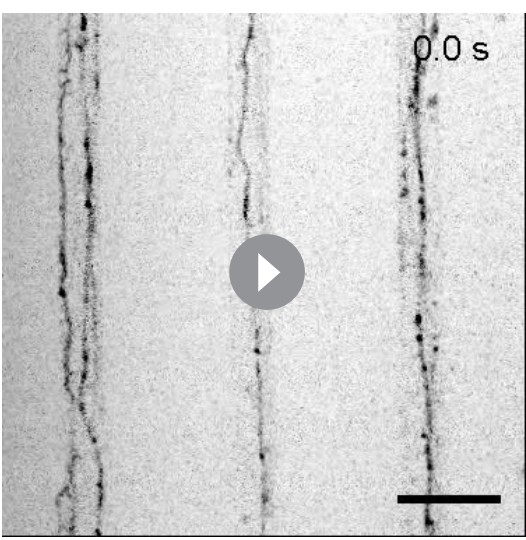

**Video 1.** Axonal Transport of APP-mCherry in presynaptic cortical neurons at DIV13. Vesicles were recorded for 60 s at 5 Hz. Axons are oriented from soma (top of the channel) to neurite terminals (bottom) with anterograde vesicles going down. Scale bar, 20 µm.

https://elifesciences.org/articles/56371#video1

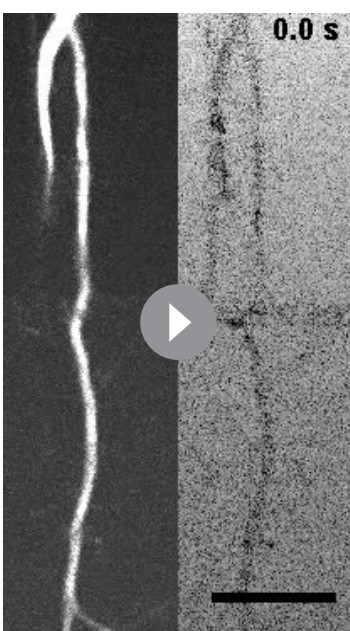

**Video 2.** Transport of APP-mCherry (right panel) in MAP2-GFP positive postsynaptic cortical dendrites (left panel) at DIV13. Vesicles were recorded for 60 s at 5 Hz. Dendrites are oriented from soma (top of the channel) to neurite terminals (bottom) with outward vesicles going down. Scale bar, 20 µm.

https://elifesciences.org/articles/56371#video2

within axons. We defined the overall direction of APP-mCherry vesicle transport in axons by adding the anterograde cumulative distance to the negative retrograde cumulative distance, so that positive values indicate a net anterograde flux from the soma towards the synapse (*Figure 2B*). We also measured dendritic transport, expressed as inward (from postsynaptic compartment to soma) or outward (from soma to postsynaptic site), since microtubules in dendrites (unlike in axons) are not fully oriented with the plus ends towards the dendrite's extremities (*Kapitein and Hoogenraad, 2015*; *van Beuningen and Hoogenraad, 2016*; *Figure 2B*). Our analysis showed a net anterograde axonal and outward dendritic flux for APP-mCherry-containing vesicles, indicating that there is a significant transport of APP to the synapse from both pre- and postsynaptic neurons (*Figure 2B*). These findings in a mature network are in accordance with the reported velocities of APP vesicles (*Fu and Holzbaur, 2013*; *Her and Goldstein, 2008*; *Marquer et al., 2014*; *Rodrigues et al., 2012*; *Vagnoni et al., 2013*).

## Huntingtin phosphorylation regulates axonal but not dendritic transport of APP

We had previously shown that phosphorylation of HTT at Serine 421 determines the direction in which various cargoes are transported in neurites (*Colin et al., 2008*). These experiments overexpressed short HTT fragments containing mutations at Serine 421 in neurons that were randomly cultured (i.e., not integrated into a mature network) and in which axons and dendrites could not be discriminated. To study the role of HTT phosphorylation at S421 in APP transport in axons *versus* dendrites, we took advantage of our microfluidic system and two lines of homozygous knock-in mice: one in which Serine 421 is replaced by an alanine ($Htt^{S421A/S421A}$ or $HTT_{SA}$), mimicking the absence of phosphorylation, and another in which Serine 421 is replaced by aspartic acid ($Htt^{S421D/S421D}$ or $HTT_{SD}$), mimicking constitutive phosphorylation (*Thion et al., 2015*). It is important to note that neither mutation affects the level of HTT expression (*Ehinger et al., 2020*).

We isolated $HTT_{SA}$ and $HTT_{SD}$ cortical neurons from the mice and plated them in both pre- and postsynaptic compartments of our microfluidic device as in *Figure 1A*. Abolishing HTT phosphorylation at Serine 421 increased the velocity of retrograde vesicles, increased their cumulative distance travelled, and reduced the net

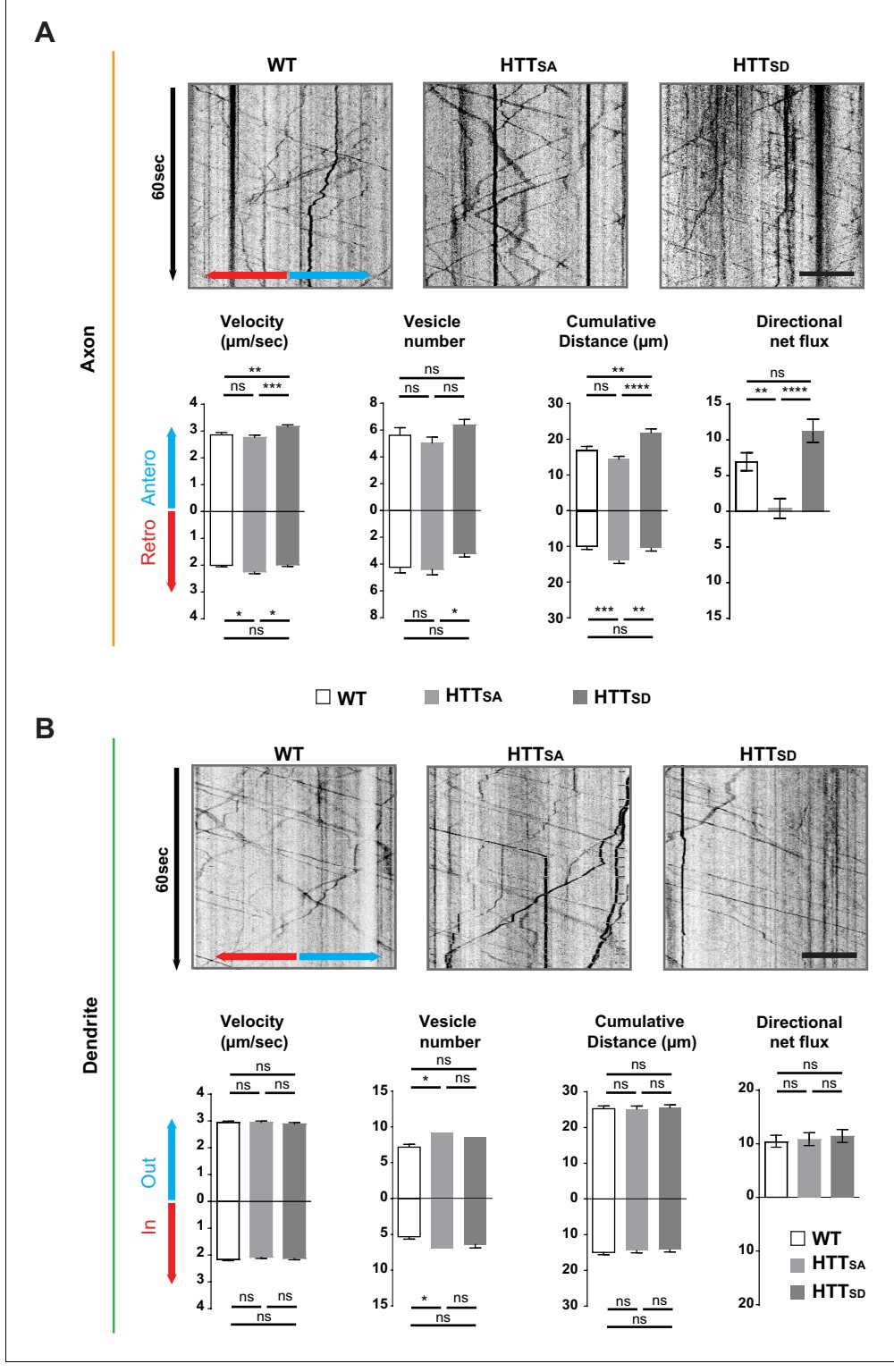

**Figure 3.** Axonal but not dendritic transport of APP depends on HTT phosphorylation. (A) Kymographs and quantifications of APP-mCherry into WT, HTT$_{SA}$ and HTT$_{SD}$ axons. Velocity, vesicle number per 100 µm of neurite length, cumulative distance and directional net flux were measured. Histograms represent means +/- SEM of 3 independent experiments, 41 WT, 52 HTT$_{SA}$ and 63 HTT$_{SD}$ axons and 674 WT, 602 HTT$_{SA}$ and 493 HTT$_{SD}$ vesicles. Significance was determined using an unpaired t-test; *p<0.05, **p<0.01, ***p<0.001, ****p<0.0001, ns = not significant. Scale bar = 20 µm. (B) Kymographs and quantifications of APP-mCherry into WT, HTT$_{SA}$ and HTT$_{SD}$

*Figure 3 continued on next page*

*Figure 3 continued*

dendrites. Dendritic inward and outward velocity, vesicle number per 100 μm of neurite length, cumulative distance and directional net flux were measured. Histograms represent means +/- SEM of 4 independent experiments, 122 WT, 99 HTT$_{SA}$ and 109 HTT$_{SD}$ dendrites, 1171 WT, 1119 HTT$_{SA}$ and 1074 HTT$_{SD}$ vesicles. Significance was determined using an unpaired t-test; *p<0.05; ns = not significant. Scale bar = 20 μm. (see also *Video 3*).

The online version of this article includes the following source data and figure supplement(s) for figure 3:

**Source data 1.** Statistical analysis of APP axonal transport.
**Source data 2.** Statistical analysis of APP dendritic transport.
**Figure supplement 1.** Cellular distribution of kinesin and dynactin in WT and HTT$_{SA}$ mouse brains.
**Figure supplement 1—source data 1.** Statistical analysis of total KHC levels.
**Figure supplement 1—source data 2.** Statistical analysis of vesicular and cytosolic KHC levels.

anterograde flux of APP vesicles in axons (*Figure 3A*, *Video 3*), whereas HTT$_{SD}$ neurons showed an increase in anterograde velocity and greater cumulative distance travelled by APP compared to HTT$_{SA}$ or wild type (WT) neurons. Nevertheless, the net flux, which reflects the flow of vesicles from the soma to axon terminals, was not significantly different from that observed in WT neurons. This indicates that in our experimental conditions, most of the WT HTT is in its phosphorylated form. Phosphorylation status did not, however, modify APP transport in dendrites (*Figure 3B*). We conclude that HTT phosphorylation regulates axonal but not dendritic transport of APP to the synapse.

Given that microtubule polarity influences selective cargo trafficking in axons and dendrites (*van Beuningen and Hoogenraad, 2016*), it is interesting to note that the axon-specific effect of HTT phosphorylation correlates with axons' preferential plus-end microtubule orientation (dendrites have mixed microtubule polarity). To further understand the selective effect in axons versus dendrites, we investigated the interaction of non-phospho HTT with kinesin-1, the molecular motor responsible for the transport of APP (*Matsuda et al., 2001*; *Verhey et al., 2001*). Because most WT HTT is already in its phosphorylated form in our experimental conditions, we compared HTT$_{SA}$ with WT (rather than HTT$_{SD}$) neurons. We found no difference between WT and HTT$_{SA}$ neurons in their total kinesin heavy chain (KHC) levels (*Figure 3—figure supplement 1A*), but HTT$_{SA}$ neurons had less KHC in the vesicular fraction than in the cytosolic fraction (*Figure 3—figure supplement 1B*). These results are in agreement with our previous study suggesting HTT dephosphorylation decreases the association of kinesin-1 with vesicles (*Colin et al., 2008*).

## HTT regulation of APP anterograde axonal transport is mediated by akt phosphorylation

HTT phosphorylation at S421 depends on the Akt kinase (*Humbert et al., 2002*). We therefore investigated whether Akt could modify anterograde transport of APP and whether this required HTT phosphorylation. We transduced cortical neurons with APP-mCherry and a construct encoding constitutively active Akt (Akt-CA) or a form of Akt (Akt-N) that has no kinase activity with and IRES-GFP or the corresponding empty GFP vector (GFP) as a control. As expected, Akt induced endogenous HTT phosphorylation in WT neurons but was unable to do so in HTT$_{SA}$ neurons (*Figure 4A*). In addition,

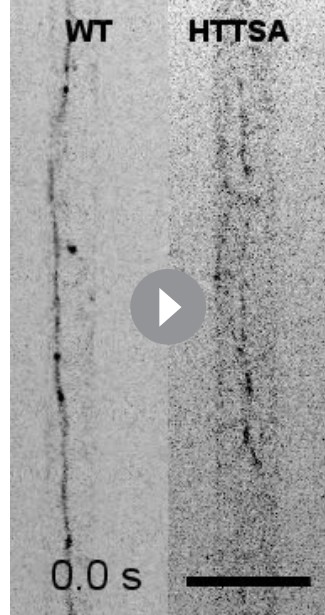

**Video 3.** APP-mCherry transport in WT (left panel) or HTT$_{SA}$ (right panel) axons at DIV13. Vesicles were recorded for 60 s at 5 Hz. Axons are oriented from soma (top of the channel) to neurite terminals (bottom) with anterograde vesicles going down. Scale bar, 20 μm.
https://elifesciences.org/articles/56371#video3

HTT phosphorylation was reduced upon Akt-N expression.

To ensure Akt was not affecting neuronal growth and maturation, we transduced neurons at DIV8 and analyzed APP trafficking at DIV13. Expressing Akt-CA and APP-mCherry in WT cortical neurons had no effect on vesicle number but markedly increased anterograde velocity of APP and the cumulative distances travelled by anterograde APP-mCherry-containing vesicles, which in turn led to an increase of their net anterograde flux (*Figure 4B and C*, *Video 4*). In contrast, in HTT$_{SA}$ neurons, Akt-CA was unable to modify the different transport parameters. Thus, Akt activation increases APP anterograde transport in axons by phosphorylating HTT at Serine 421. These results identify Akt-HTT signaling as a new mechanism that regulates axonal trafficking of APP.

## Huntingtin-mediated axonal transport determines presynaptic APP levels

To determine whether reduced anterograde axonal transport of APP affects the targeting of APP at the plasma membrane, we used TIRF (total internal reflection fluorescence) microscopy and a super-ecliptic version of pHluorin (SEP) fused to the N-terminal part of APP to monitor insertion of APP into the plasma membrane. As we could not reliably detect APP-SEP at the membrane in primary cultures of neurons, we transfected the APP-SEP construct with versions of full-length wild type HTT (pARIS HTT) (*Pardo et al., 2010*) or full-length HTT containing the S421A mutation (pARIS HTT$_{SA}$) into COS cells that are known to have their plus-end microtubules oriented toward the plasma membrane (*Takemura et al., 1995*). We detected far fewer APP-SEP dots per minute in cells expressing pARIS HTT$_{SA}$ than in cells expressing pARIS HTT (*Figure 5A*, *Video 5*). This result suggests that reducing transport of APP to the plasma membrane by dephosphorylating HTT decreases APP targeting at the plasma membrane.

The fluidic isolation of the synaptic compartment enabled us to collect proteins and investigate the targeting of APP at synapses by measuring APP levels by western blot. We first verified that the synaptic chamber is enriched with synaptic marker synaptophysin and empty of nuclear marker lamin B1 (*Figure 5B*). Lack of HTT phosphorylation led to a reduction of APP protein levels at synapses but no real change in the soma-containing chamber (*Figure 5B*).

We then investigated APP targeting in vivo. We first prepared synaptosomal fractions from WT mouse brains and purified post-synaptic density fractions (PSD, enriched in postsynaptic proteins) and non-PSD fractions that are enriched with presynaptic proteins. As expected, we detected synaptophysin, a presynaptic marker, and PSD95, a postsynaptic marker, in the non-PSD and PSD enriched fractions, respectively (*Figure 5C*). We detected APP in both fractions. We found that most of the synaptosomal APP was enriched in the non-PSD fraction, which suggests that a significant fraction of APP found at synapses originates from the presynaptic compartment. Since anterograde axonal transport of APP is controlled by HTT phosphorylation, we measured APP within fractions prepared from HTT$_{SA}$ homozygous mouse brains (*Figure 5D*). APP levels were lower in the non-PSD fraction of HTT$_{SA}$ mouse brains (enriched with presynaptic proteins), but APP levels did not differ significantly between WT and HTT$_{SA}$ mouse brains in the PSD fraction (*Figure 5D*). Together, our results indicate that the absence of HTT phosphorylation reduces anterograde transport of APP in axons, but not in dendrites, and subsequently regulates the levels of APP in the presynaptic compartment both in vitro and in vivo.

## HTT chronic dephosphorylation alters brain morphology and synapse size and number

The previously generated HTT$_{SA}$ mice have no obvious phenotype but were not fully characterized for brain-related behavior and morphology (*Thion et al., 2015*). Subsequent analyses of the mice at 6 months of age did not reveal any behavioral abnormalities (*Ehinger et al., 2020*). Given our observation that anterograde transport and presynaptic accumulation of APP are both reduced in HTT$_{SA}$ mice and that APP is associated with late-onset defects, we subjected the mice to complete behavioral analysis (SHIRPA *Rogers et al., 1997*, open field, grip test and elevated plus Maze) when they were 12 months old, and again found no significant differences in behavior (*Figure 6—figure supplement 1* and *Supplementary file 1*).

We then performed anatomical ex vivo MRI of young adult WT and HTT$_{SA}$ mice (*Figure 6A and B*). HTT$_{SA}$ mice between 8 and 11 weeks of age showed greater whole brain volume (4.8%) affecting

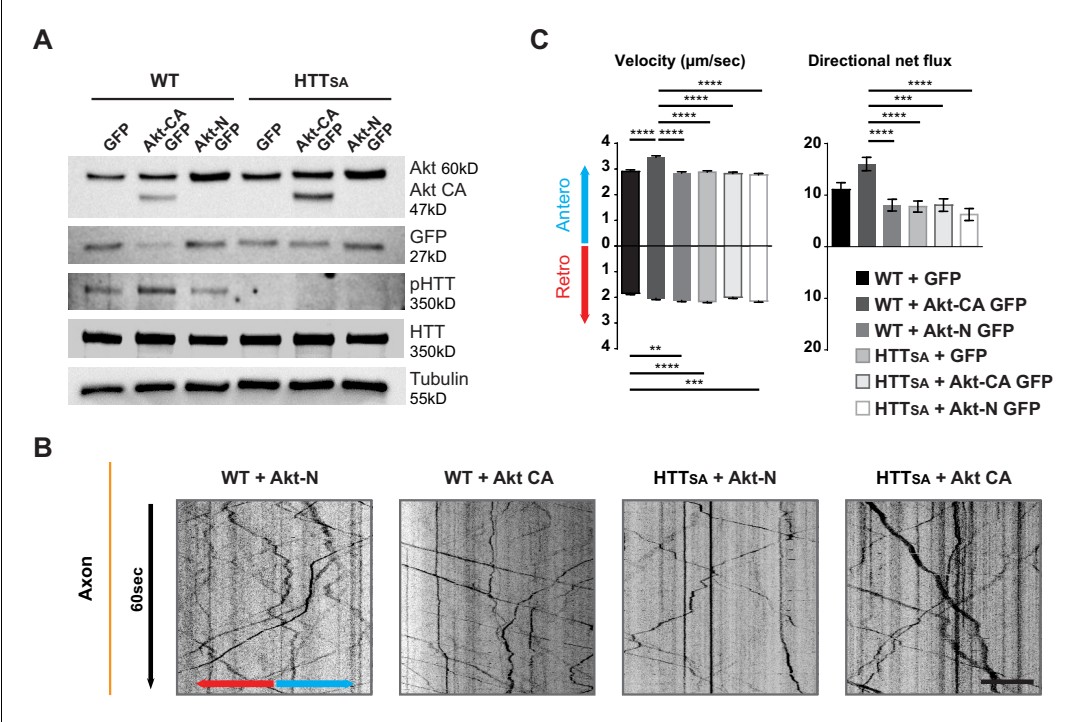

**Figure 4.** Akt regulates APP transport in a HTT phospho-dependent manner. (A) WT and HTT$_{SA}$ neurons transduced with constitutively active Akt (Akt-CA GFP) or an inactive form of Akt (Akt-N GFP) in IRES GFP constructs or with empty GFP vector (GFP) were analyzed by western blotting with Akt, GFP, phosphorylated HTT, total HTT and tubulin antibodies. (B) Kymographs of APP-mCherry from WT and HTT$_{SA}$ neurons seeded in microchambers and transduced with APP-mCherry and GFP, Akt-CA GFP or Akt-N GFP. Scale bar 20 μm. (see also *Video 4*). (C) Velocity and directional net flux of APP-mCherry vesicles were quantified. Histograms represent means +/- SEM of 3 independent experiments, 936 WT GFP, 988 WT AKT CA, 1261 WT AKT N, 1357 HTT$_{SA}$ GFP, 1048 HTT$_{SA}$ AKT CA and 1177 HTT$_{SA}$ AKT N vesicles. Significance was determined using one-way ANOVA followed by Tukey's post-hoc analysis for multiple comparisons; *p<0.05, **p<0.01, ***p<0.001, ****p<0.0001.

The online version of this article includes the following source data for figure 4:

**Source data 1.** Statistical analysis of APP axonal transport according to Akt activity.

the hippocampus (8.5%) and the cortex (3.7%) but not the striatum. To determine whether HTT$_{SA}$ produced more subtle changes in synapse number and morphology, we quantified synaptic density and spine size in WT and HTT$_{SA}$ mice by electron microscopy. HTT$_{SA}$ mice had more synapses than WT mice but no difference in spine size (*Figure 6C*).

To further investigate the contribution of transport and APP levels on synapse number, we took advantage of the microfluidic devices and, using pre- and post-synaptic markers (synaptophysin and PSD95, respectively), measured the number of synaptic contacts in the synaptic compartment within WT and HTT$_{SA}$ mature neuronal circuits at DIV12. In agreement with our in vivo experiments (*Figure 6C*), we found an increase in the number of synaptic contacts in the HTT$_{SA}$ circuit (*Figure 7A*).

To determine whether APP overexpression would increase the quantity of APP within the

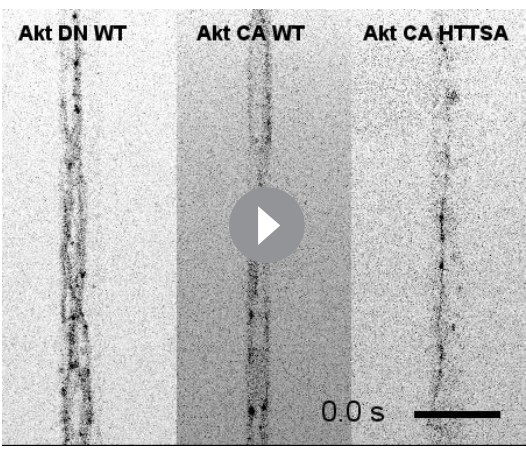

**Video 4.** Effect of Akt on the axonal transport of APP-mCherry in presynaptic cortical neurons from WT or HTT$_{SA}$ mice at DIV13. Vesicles were recorded for 60 s at 5 Hz. Axons are oriented from soma (top of the channel) to neurite terminals (bottom) with anterograde vesicles going down. Scale bar, 20 μm.

https://elifesciences.org/articles/56371#video4

presynaptic cortical compartment of a WT or HTT$_{SA}$ network (*Figure 7B*), we transduced WT or HTT$_{SA}$ neurons with a lentivirus expressing APP-mCherry at DIV7 and measured synapse number at DIV12. Overexpressing APP-mCherry in WT presynaptic cortical neurons decreased synaptic contacts, but overexpressing APP-mCherry in HTT$_{SA}$ presynaptic cortical neurons restored synaptic contacts back to the levels seen in WT neurons (*Figure 7B*). The presynaptic level of APP thus appears to determine synapse number and can be modulated by HTT phosphorylation; this further supports a role for the Akt-HTT-APP pathway in synapse homeostasis. To ensure that the HTT$_{SA}$ mutation was not affecting neurodevelopment, we transduced a WT circuit at DIV8, when axon growth has ended (*Moutaux et al., 2018*), with lentiviruses expressing APP-mCherry and either an N-terminal HTT construct containing the first 480 amino acids (HTT-480-WT) or a construct in which the S421 has been mutated into alanine (HTT-480-SA). We found that expressing the HTT-480-SA construct in mature neurons led to an increase in synaptic contacts similar to what is observed in HTT$_{SA}$ neurons differentiated in microchambers (*Figure 7A and C*). This suggests that the HTT S421A mutation has no major role in axon growth and/or that the increase of synaptic contacts seen in HTT$_{SA}$ neurons is not due to changes in neurodevelopment but rather results from reduced transport and accumulation of APP at the presynapses. We then investigated the effect of APP overexpression in WT neurons. As in *Figure 7B*, APP-mCherry overexpression in WT neurons transduced with HTT-480-WT led to a decrease in the number of synaptic contacts. However, it had no effect in neurons expressing HTT-480-SA, indicating that HTT dephosphorylation attenuates the effect of APP overexpression on synapse number (*Figure 7D*). We conclude that reducing anterograde axonal transport of APP either during axonal growth or in mature networks is sufficient to modulate synaptic contact number.

## Unphosphorylatable HTT reduces APP presynaptic levels in APPPS1 mouse model

HTT-mediated transport clearly modulates presynaptic levels of APP in a corticocortical circuit, but we wanted to investigate the consequences of chronic HTT dephosphorylation in vivo, in a mouse line that overexpresses APP. We chose APPPS1 mice—double transgenics that bear a human APP transgene with the Swedish mutation (APP$^{Swe}$) and a mutant human presenilin 1 (PS1$^{L166P}$) transgene (*Radde et al., 2006*)—which express human APP at three times the level of murine APP and mimic familial Alzheimer's. These mice show reduced synapse density (*Alonso-Nanclares et al., 2013*; *Bittner et al., 2012*; *Hoe et al., 2012*; *Müller et al., 2017*; *Priller et al., 2009*; *Radde et al., 2006*; *Zou et al., 2015*). We crossed HTT$_{SA}$ phospho-mutant mice with APPPS1 mice to generate *APPswe; PS1L166P;Htt*$^{S421A/S421A}$ mice, heretofore referred to as APPPS1/HTT$_{SA}$ mice.

The levels of APP in the non-post-synaptic density fraction of APPPS1/HTT$_{SA}$ mice were significantly lower than in APPPS1 mice at 10 months (*Figure 8A and B*). Since the PS1$^{L166P}$ mutation promotes APP cleavage, thereby increasing Aβ42 production (*Radde et al., 2006*), we biochemically quantified Aβ levels (*Figure 8—figure supplement 1A*) and performed histological measurements of plaque loads using the 4G8 antibody that recognizes both human and murine Aβ (*Figure 8—figure supplement 1B*). We also measured amyloid burden using Congo red to stain amyloid plaques, and OC and A11 antibodies to recognize amyloid fibrils, fibrillary oligomers, and prefibrillar oligomers (*Kayed et al., 2007*) in 19-month-old APPPS1 and APPPS1/HTT$_{SA}$ mouse brains, a time that corresponds to the final behavioral evaluation of the mice before histopathological analyses (*Figure 8—figure supplement 1C*). We found no significant differences between genotypes, indicating that loss of HTT phosphorylation has no effect on Aβ level, amyloid load, Aβ oligomer load or plaque aggregation. These results indicate that HTT dephosphorylation regulates presynaptic levels of APP$^{Swe}$ without affecting downstream Aβ production and/or accumulation.

We next quantified the number and the size of the spines in the CA1 region of 19-month-old mice by electron microscopy. As previously described for APP$^{Swe}$-PS1$^{ΔE9}$ mice, 5xFAD Tg mice, and APPxPS1-KI mice (*Androuin et al., 2018*; *Koffie et al., 2009*; *Neuman et al., 2015*), APPPS1 mice showed lower synaptic density and larger spines than WT mice (*Figure 8C*). Strikingly, unphosphorylatable HTT (HTT$_{SA}$) significantly increased spine density and completely rescued APPPS1-induced increase of spine size (*Figure 8C*).

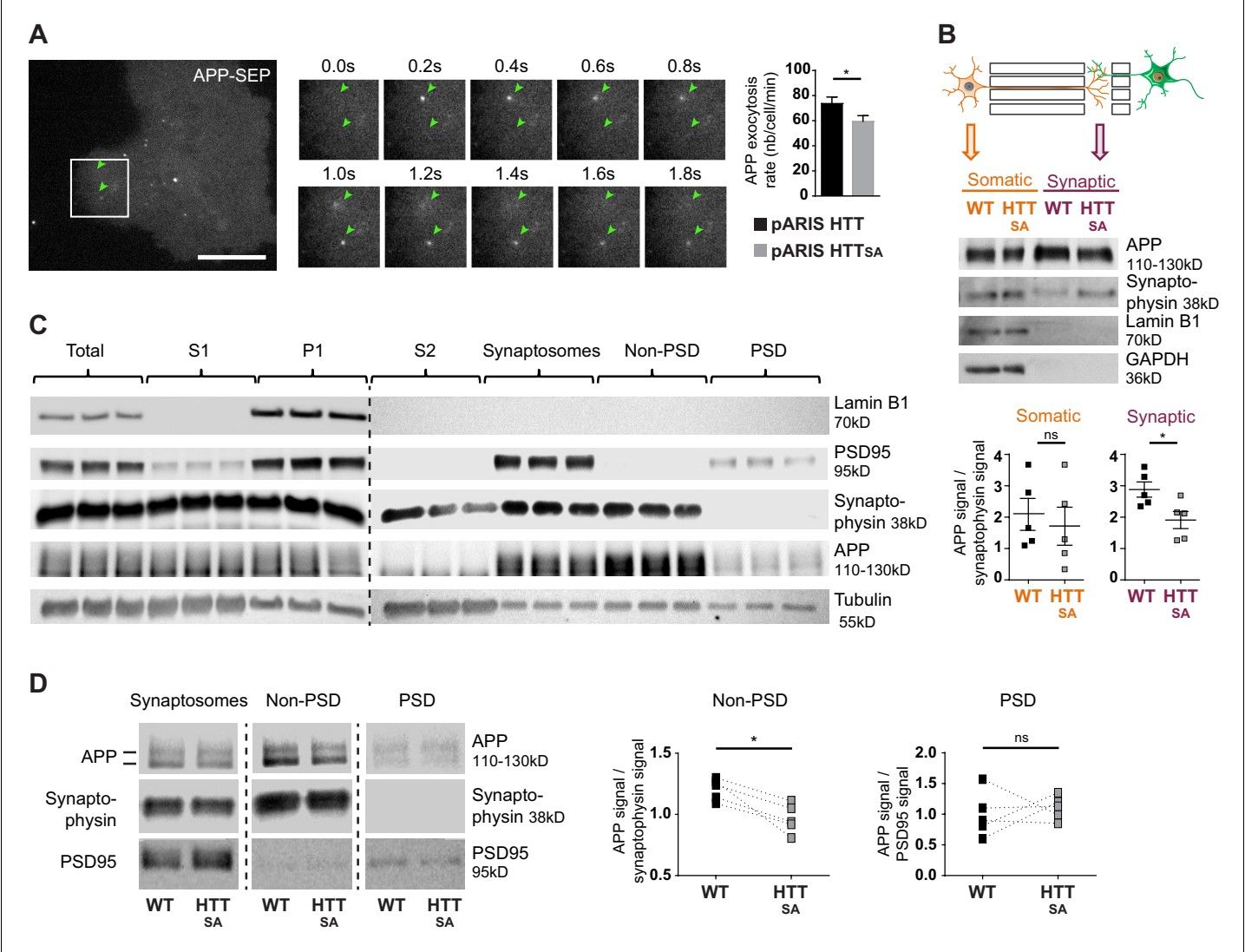

**Figure 5.** HTT S421 phosphorylation affects presynaptic APP targeting. (**A**) Effect of HTT S421 phosphorylation on exocytosis rate of APP was analyzed in COS cells co-transfected with APP-SEP (Super Ecliptic pHluorin) and with pARIS HTT or pARIS HTT$_{SA}$ visualized by TIRF microscopy. Magnification represents a time lapse of events showing 2 events of APP vesicle exocytosis (green arrows). Histograms represent means +/- SEM of exocytosis event number per minute in 39 HTT and 40 HTT$_{SA}$ cells from four independent experiments. Significance was determined using an unpaired t-test; *p<0.05. Scale bar = 20 μm. (see also **Video 5**). (**B**) Effect of HTT S421 phosphorylation on APP targeting at the synapse was assessed by anti-APP western blotting (22C11) analysis of extracts from synaptic chambers of a WT or HTT$_{SA}$ corticocortical network. SNAP25 was used as a control for protein content in the synaptic compartment and nuclear marker Lamin B1 for the somatic compartment. Histograms represent means +/- SEM of APP signal per synaptophysin signal on five independent experiments. Significance was determined using a Mann-Whitney test; *p<0.05, ns = not significant. (**C**) Western blotting analysis of pre- and postsynaptic fractions obtained from synaptosome preparations. Fractionation gives the first pellet, P1, the first supernatant, S1, and the second supernatant, S2. Lamin B1, a nuclear marker is enriched in P1 fraction. The pre- (non-PSD) and the post-synaptic (PSD) fractions are respectively enriched in synaptophysin and PSD95. (**D**) APP from WT or HTT$_{SA}$ cortices fractions was quantified by western blotting analyses. APP signal was quantified as the ratio of synaptophysin signal for non-PSD fraction and as the ratio of PSD95 signal for PSD fraction. One line represents one experiment. Significance was determined using Mann-Whitney test; *p<0.05, ns = not significant.

The online version of this article includes the following source data for figure 5:

**Source data 1.** Statistical analysis of APP exocytosis rate.
**Source data 2.** Statistical analysis of APP levels in microfluidics device.
**Source data 3.** Statistical analysis of APP levels in synaptosome from brains.

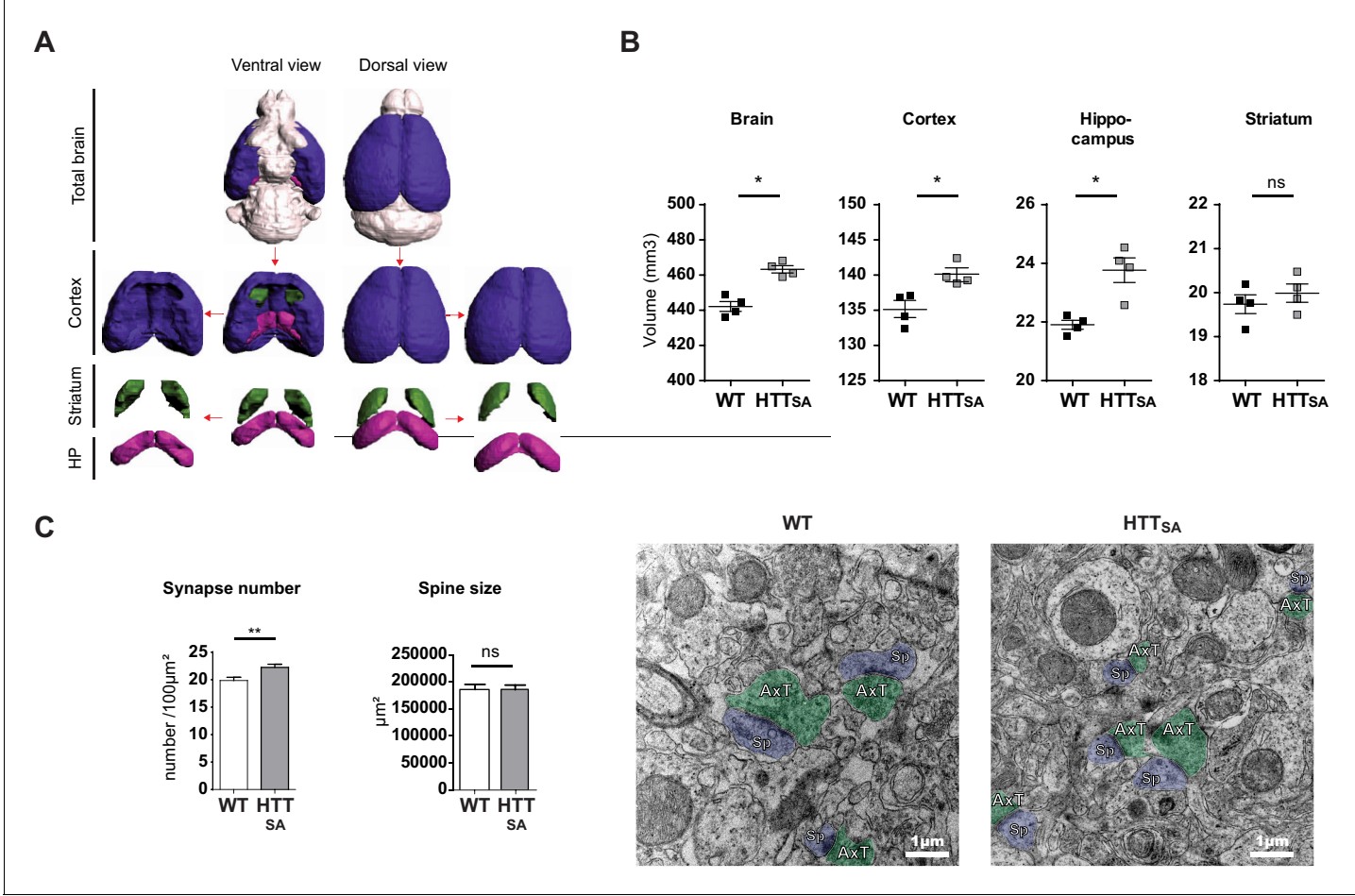

**Figure 6.** HTT dephosphorylation induces changes in brain morphology and synapse number. (**A**) Representative 3D reconstructions of WT brain areas built from high spatial resolution ex vivo MRI-T$_{1w}$ data. Each brain structure is represented with a specific color: cortex (purple), hippocampal formation (pink), striatum (green), and other structures (light grey). (**B**) Quantification of the volumes of the different cerebral regions represented in (**A**). The graphics show volumes for these regions (in mm$^3$) of 2 and 3-month-old WT and HTT$_{SA}$ mice. Black bars represent the mean of 4 WT and 4 HTT$_{SA}$ mice, Mann and Whitney two tails, $^*P < 0.05$; $^{**}P < 0.01$; $^{***}P < 0.001$. (**C**) Synapse number and size in CA1 region of 19-month-old WT or HTT$_{SA}$ mice were quantified by electron microscopy. Axon terminals (AxT) and spines (Sp) are colored with green and purple respectively. Histograms represent means +/- SEM of 3 brains with 134 (WT) and 203(HTT$_{SA}$) fields analyzed and 225 (WT) and 218 (HTT$_{SA}$) synapses. Significance was determined using an unpaired t-test; **p<0.01, ns = not significant. Scale bar = 1 µm.

The online version of this article includes the following source data and figure supplement(s) for figure 6:

**Source data 1.** Statistical analysis of brain structure volumes of HTTSA mice.
**Source data 2.** Statistical analysis of synapse number and spine size in HTTSA brains.
**Figure supplement 1.** Behavioral analyses of the locomotion, force and anxiety in WT and HTT$_{SA}$ mice.
**Figure supplement 1—source data 1.** Statistical analysis of the distance moved in an open field by HTTSA mice.
**Figure supplement 1—source data 2.** Statistical analysis of the time spent in the periphery of an open field by HTTSA mice.
**Figure supplement 1—source data 3.** Statistical analysis of grip force test by HTTSA mice.
**Figure supplement 1—source data 4.** Statistical analysis of EPM test by HTTSA mice.

## Unphosphorylatable HTT improves learning and memory in APPPS1 mice

We next investigated the behavior of WT, APPPS1, and APPPS1/HTT$_{SA}$ mice. Extensive phenotypic analysis of the HTT$_{SA}$ mice using a modified SHIRPA primary screen and various behavioral tests (*Figure 6—figure supplement 1* and *Supplementary file 1*) revealed no significant changes in the behavioral-neurological status of 12 to 15 month-old HTT$_{SA}$ mice compared to WT mice. When we

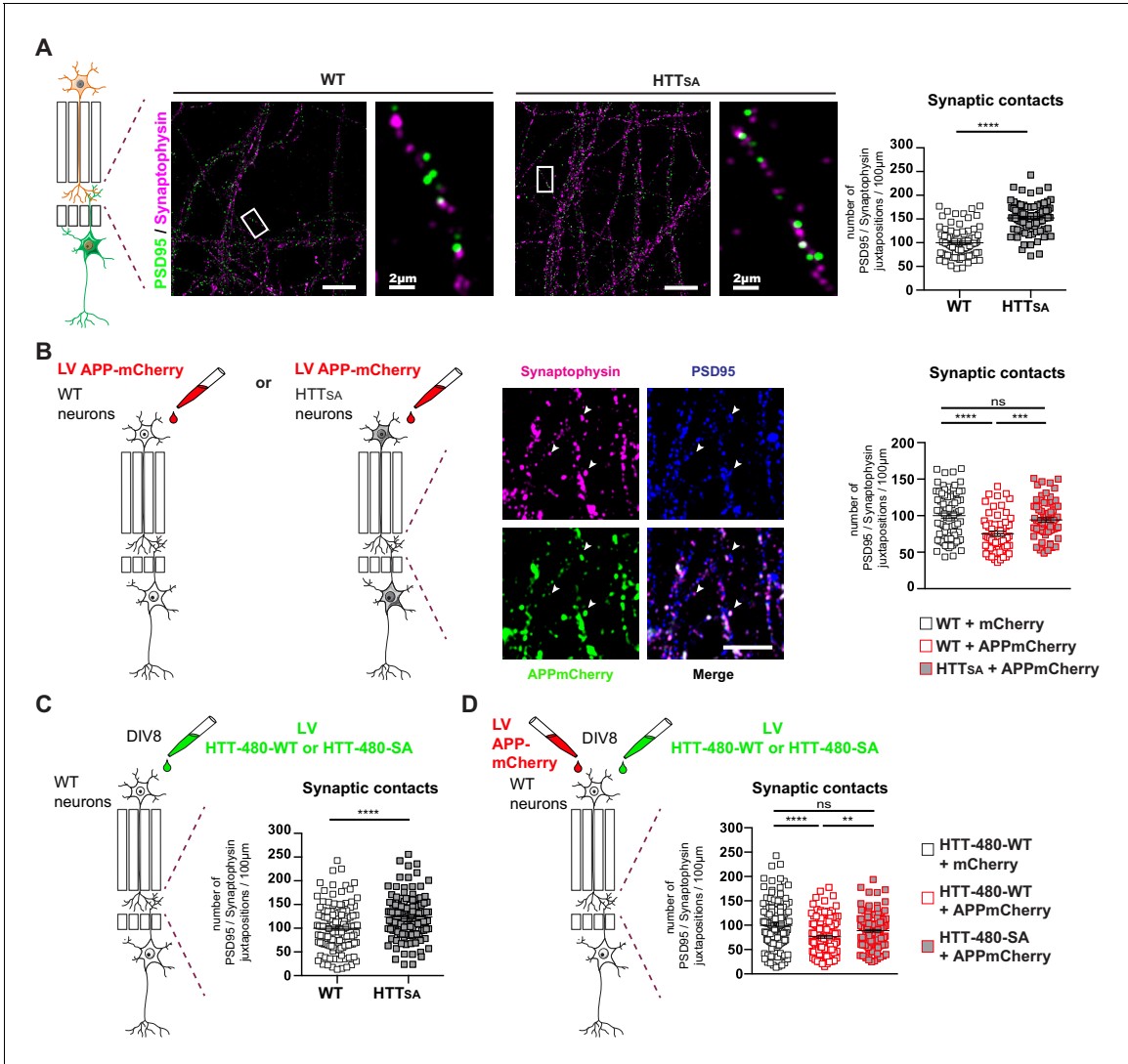

**Figure 7.** HTT phosphorylation regulates synaptic contacts by reducing presynaptic APP levels. (**A**) Number of PSD95/Synaptophysin contacts in the synaptic chamber of WT and HTT$_{SA}$ network. Right microphotographs for each genotype show magnification of representative neurites. Scale bars = 20 µm (low magnification) or 2 µm (high magnification). Histograms represent means +/- SEM of 3 independent experiments and 85 WT and 91 HTT$_{SA}$ neurites. Significance was determined using an unpaired t-test; ****p<0.0001. (**B**) Representative image of APP-mCherry transduced presynaptic neurons. APP-mCherry is present in axon terminals positive for synaptophysin (white arrows). Scale bar = 2 µm. Number of PSD95/Synaptophysin contacts in the synaptic chamber of WT and HTT$_{SA}$ network transduced at presynaptic site with APP-mCherry or mCherry as a control. Histograms represent means +/- SEM of 3 independent experiments and 75 WT + mCherry; 59 WT + APP-mCherry and 71 HTT$_{SA}$ APP-mCherry neurites. (**C**) Number of PSD95/Synaptophysin contacts in the synaptic chamber of WT mature network transduced at presynaptic site with a lentivirus encoding an HTT construct containing the first 480 amino acids without (HTT-480-WT) or with the S421A mutation (HTT-480-SA). Histograms represent means +/- SEM of at least three independent experiments and 132 HTT-480-WT and 130 HTT-480-SA neurites. Significance was determined using Mann and Whitney test; ****p<0.0001. (**D**) Number of PSD95/Synaptophysin contacts in the synaptic chamber of WT mature network transduced at presynaptic site with APP-mCherry or mCherry as a control and with a lentivirus encoding a HTT-480-WT or HTT-480-SA. Histograms represent means +/- SEM of 3 independent experiments and 132 HTT-480-WT + mCherry, 134 HTT-480-WT + APP mCherry and 136 HTT-480-SA + APP mCherry neurites. Significance was determined using one-way Kruskal-Wallis test followed by Dunn's post-hoc analysis for multiple comparisons; **p<0.01, ***p<0.001, ****p<0.0001, ns = not significant.

The online version of this article includes the following source data for figure 7:

**Source data 1.** Statistical analysis of the number of synaptic contacts in HTTSA neurons.
**Source data 2.** Statistical analysis of the number of synaptic contacts in APP transduced neurons.
**Source data 3.** Statistical analysis of the number of synaptic contacts in HTT transduced neurons.
**Source data 4.** Statistical analysis of the number of synaptic contacts in APP and HTT transduced neurons.

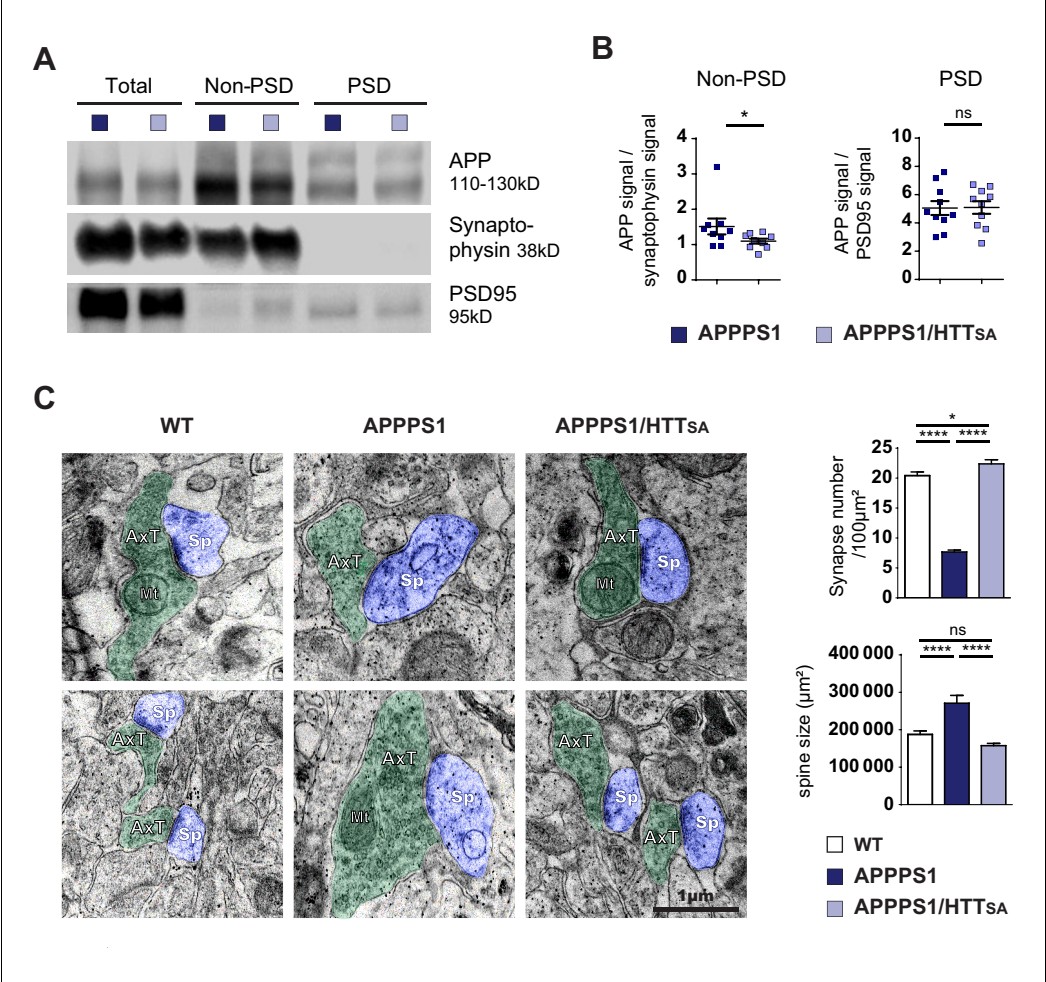

**Figure 8.** HTT S421 dephosphorylation rescues synapse number in APPPS1 mice. (**A**) APP levels from APPPS1 and APPPS1/HTT_{SA} cortical fractions were quantified by western blotting analyses after synaptosomes fractionation. (**B**) APP signal was quantified as the ratio of synaptophysin signal for non-PSD fraction and as the ratio of PSD95 signal for PSD fraction. Histograms represent means +/- SEM of 9 experiments. Significance was determined using Wilcoxon test; *p<0.05, ns = not significant. (**C**) Synaptic number and postsynaptic density (PSD) length of CA1 region of hippocampi from 19-month-old APPPS1 and APPPS1/HTT_{SA} mice were quantified by electron microscopy. Axon terminals (AxT) and spines (Sp) are colored with green and purple, respectively. Scale bar = 1 μm. (**D**) Histograms represent means +/- SEM of 3 brains; 153 APPPS1 and 152 APPPS1/HTT_{SA} fields and 182 APPPS1 and 350 APPPS1/HTT_{SA} synapses were analyzed. Significance was determined using one-way ANOVA followed by Tukey's multiple comparisons test; *p<0.05, ****p<0.0001; ns = not significant.

The online version of this article includes the following source data and figure supplement(s) for figure 8:

**Source data 1.** Statistical analysis of APP levels in synaptosomes from APPPS1/HTTSA brains.
**Source data 2.** Statistical analysis of synapse number and spine size of APPPS1/HTTSA brains.
**Figure supplement 1.** Analysis of soluble Aβ42 levels, amyloid plaques and amyloid load in APPPS1 and APPPS1/HTT_{SA} mice.
**Figure supplement 1—source data 1.** Statistical analysis of soluble of Aβ42 levels in APPPS1/HTTSA brains.
**Figure supplement 1—source data 2.** Statistical analysis of amyloïd load in APPPS1/HTTSA brains.

compared WT, APPPS1, and APPPS1/HTT_{SA} mice, we found no significant differences in locomotor activity or anxiety-related behavior in the open field test (*Figure 9—figure supplement 1*).

We then evaluated spatial learning of 12- to 15-month-old APPPS1 and APPPS1/HTT_{SA} mice in the Morris water maze paradigm. As expected, APPPS1 mice took longer paths to reach the platform of the water maze (*Figure 9A*). APPPS1/HTT_{SA} mice performed better than APPPS1 mice, although not to the level of WT mice. Comparing the early and late stages of learning in the APPPS1/HTT_{SA} mice, we found that APPPS1/HTT_{SA} mice performed about as poorly as the APPPS1 mice in the early stages of learning (first training sessions). In the late training sessions, however, they performed much better than APPPS1 mice and showed a substantial recovery of performance

(*Figure 9B*). Finally, we evaluated their memory of the platform location by subjecting the mice to a probe trial. APPPS1 mice explored all quadrants of the pool equally, whereas both WT and APPPS1/HTT$_{SA}$ mice showed a preference for the target quadrant, indicating that their memory of the platform was intact (*Figure 9C*).

We also subjected the mice to the novel object recognition test (*Figure 9D*). APPPS1 mice spent similar time investigating familiar and novel objects, indicating a memory deficit (indicated by a memory index close to 50%). In contrast, APPPS1/HTT$_{SA}$ mice behaved as WT mice and showed a marked preference for the novel object (significantly different from the theoretical 50% random score), suggesting that unphosphorylatable HTT can mitigate the memory deficit observed in APPPS1 mice.

We conclude that blocking Akt phosphorylation at HTT S421 reduces APP presynaptic levels, improving learning and memory in APPPS1 mice.

## Discussion

We used high-resolution live-cell imaging of isolated axonal and dendritic compartments in a mature corticocortical network-on-a-chip to investigate the influence of HTT phosphorylation on APP

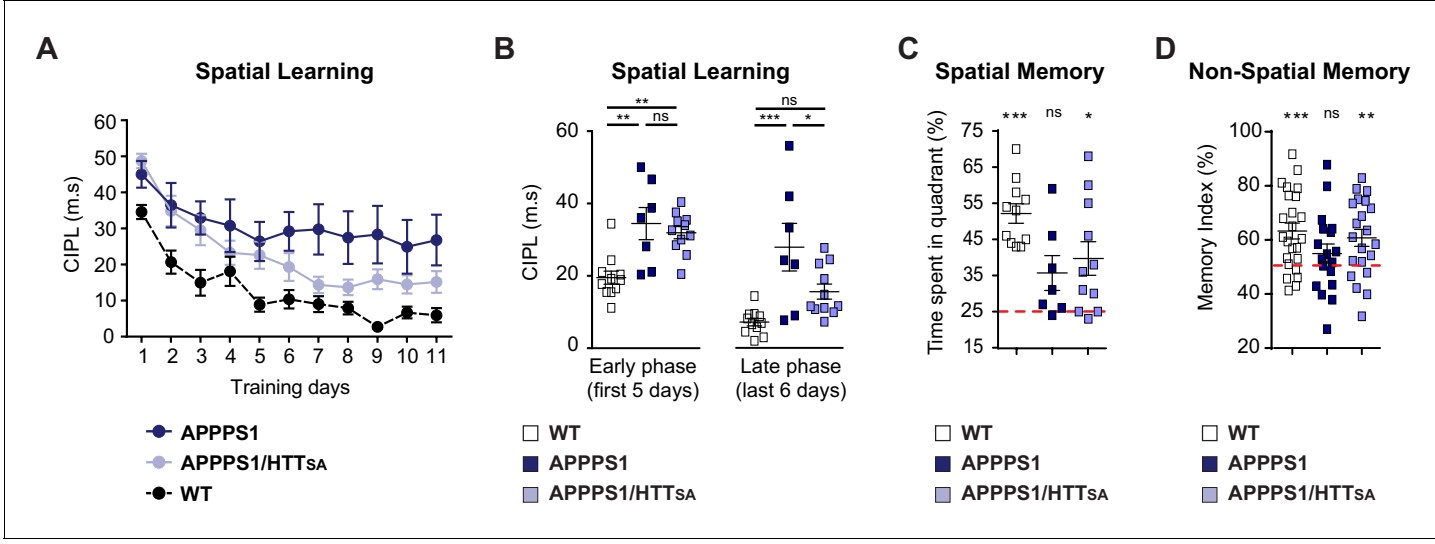

**Figure 9.** HTT S421 dephosphorylation enhances learning and memory in APPPS1 mice. (**A**) Spatial learning of 7 APPPS1 mice (dark blue), 11 APPPS1/HTT$_{SA}$ mice (light blue), and 11 WT mice (black dotted line) was assessed by measuring CIPL (Corrected Integrated Path Length), an unbiased measure of learning in the Morris water maze test over 11 days of training. Data are represented as mean ± SEM. (**B**) Cumulative CIPL during the early phase (first 5 days) and the late phase (last 6 days) of training is depicted for WT, APPPS1 and APPPS1/HTT$_{SA}$ mice. All values are means ± SEM. Significance was determined using one-way ANOVA test followed by Tukey's post-hoc analysis for multiple comparisons; *p<0.05, **p<0.01, ***p<0.001; ns = not significant. (**C**) Spatial memory of 11 WT, 7 APPPS1 and 11 APPPS1/HTT$_{SA}$ mice was assessed on a probe trial performed 72 hr after the last training day and during which the percentage of time spent in the target quadrant was quantified. All values are means ± SEM. Significance above the 25% chance level was determined using a one-sample t-test for each group. *p<0.05, ***p<0.001; ns = not significant. (**D**) Non-spatial memory of 24 WT, 18 APPPS1 and 21 APPPS1/HTT$_{SA}$ mice was assessed by the novel object recognition memory test. Memory index is calculated as the percentage of time spent exploring a novel object versus the time spent exploring both familiar and novel objects after a retention interval of 3 hr. All values are means ± SEM. A score of 50% indicates no preference (i.e., no memory). Performance significantly above the 50% chance level was determined using a one-sample t-test for each group. **p<0.01, ***p<0.001; ns = not significant.

The online version of this article includes the following source data and figure supplement(s) for figure 9:

**Source data 1.** Statistical analysis of spatial learning of APPPS1/HTTSA mice.

**Source data 2.** Statistical analysis of cumulative CIPL.

**Source data 3.** Statistical analysis of spatial memory of APPPS1/HTTSA mice.

**Source data 4.** Statistical analysis of non spatial memory of APPPS1/HTTSA mice.

**Figure supplement 1.** Behavioral analyses of the locomotion and anxiety-related behavior in WT, APPPS1 and APPPS1/HTT$_{SA}$ mice.

**Figure supplement 1—source data 1.** Statistical analysis of the distance moved in an open field by APPPS1/HTTSA mice.

**Figure supplement 1—source data 2.** Statistical analysis of the time spent in the periphery of an open field by APPPS1/HTTSA mice.

trafficking. We then evaluated the consequences of HTT phosphorylation on brain morphology and function in both wild-type mice and in transgenic mice with AD-like neuropathology. We propose a model in which axonal transport of APP, APP presynaptic levels, and synapse homeostasis require an intact Akt-HTT pathway.

## HTT links APP axonal transport to presynaptic levels of APP

We find that axonal, but not dendritic, transport of APP is regulated by Akt-phosphorylated HTT. We previously showed that HTT phosphorylation at Serine 421 recruits kinesin-1 to the molecular motor complex and promotes anterograde transport of vesicles to the plus end of microtubules in axons (*Colin et al., 2008*). Given the mixed polarity of microtubules in dendrites (*Kapitein and Hoogenraad, 2015*; *Yau et al., 2016*), modifying HTT phosphorylation affects APP transport only in axons, where all the microtubules are oriented with the plus end towards the axon terminal. Our finding that the modulation of axonal transport of APP regulates synaptic APP homeostasis is in agreement with nerve ligation studies, which showed early on that blocking traffic from the entorhinal cortex to the dentate gyrus greatly reduced APP levels at the synapse (*Koo et al., 1990*). Results obtained by pulse-chase labeling experiments and unilateral lesions of the perforant path, a circuit by which axons from the entorhinal cortex connect to the dentate gyrus, also accord with these results (*Buxbaum et al., 1998*; *Lazarov et al., 2002*). Moreover, studies investigating the composition of APP vesicles report that most proteins co-transported with APP are presynaptic (*Kohli et al., 2012*; *Szodorai et al., 2009*) and that APP colocalizes with presynaptic proteins at the presynaptic bouton (*Groemer et al., 2011*). Notwithstanding the contribution of dendritic APP to synapse homeostasis (*Niederst et al., 2015*), our results demonstrate that APP levels at presynaptic membranes rely on HTT-dependent axonal transport.

## HTT phosphorylation, APP presynaptic levels and synapse homeostasis

The absence of HTT phosphorylation reduced presynaptic levels of APP, restored synapse number and PSD length, and attenuated memory deficits in APPPS1 mice. In contrast, there was no effect of the HTT S421A mutation on amyloid plaques, on different pools of Aβ oligomers, or on extracellular and intracellular Aβ42 levels in brain, although the existence of a pool of vesicular presynaptic Aβ has been recently reported (*Yu et al., 2018*). These results suggest that HTT dephosphorylation regulates synapse homeostasis by modulating presynaptic APP levels rather than modulating APP-derived Aβ production. Notwithstanding the synaptic toxicity of Aβ peptides (*Klementieva et al., 2017*; *Mucke and Selkoe, 2012*; *Palop and Mucke, 2010*; *Selkoe and Hardy, 2016*; *Wei et al., 2010*), our findings dovetail nicely with previous results showing presynaptic APP contributes to synapse formation, function, and maintenance (*Hoe et al., 2012*; *Müller et al., 2017*; *Nicolas and Hassan, 2014*).

Our results are also in accord with reports that *App* knockout increases the number of functional synapses in vitro (*Priller et al., 2006*) and augments synaptic density in vivo, as visualized by two-photon in vivo microscopy through a cranial window (*Bittner et al., 2009*). Indeed, we found that reducing APP presynaptic levels by blocking HTT phosphorylation increases synapse density, and that this effect can be reversed by over-expressing APP in presynaptic neurons. Conversely, and again in agreement with previous studies (*Alonso-Nanclares et al., 2013*; *Bittner et al., 2012*; *Priller et al., 2009*), we found that overexpressing APP and PS1 mutations in vivo reduced synaptic density, an effect that can be restored by HTT-mediated reduction of APP presynaptic levels, without having significant effects on Aβ levels. Although the precise physiological function of APP at the synapse remains to be elucidated, changes in synapse homeostasis could be linked to the potential function of APP as an adhesion molecule that forms homo and/or heteromeric complexes with APP family members (*Müller et al., 2017*; *Soba et al., 2005*).

## Relevance to disease pathogenesis

Our identification of an Akt-HTT pathway that regulates APP and synapse homeostasis might be of relevance for AD pathophysiology. Post-mortem analyses of AD patient brains report increased levels of activated Akt in mid-temporal and mid-frontal cortex soluble fractions (*Rickle et al., 2004*) as well as increased phosphorylation of Akt and of Akt substrates in membrane-bound fractions (*Griffin et al., 2005*). Our finding that inhibiting Akt-mediated HTT phosphorylation reduces APP

presynaptic levels in APPPS1 mice suggests that increased Akt activity might contribute to higher presynaptic APP levels in AD brains, leading to synapse loss and cognitive decline. Our findings may also relate to studies that show rescue of synaptic and behavioral deficits in AD mouse models by knocking down the IGF-1 receptor and inhibiting the phosphoinositide three kinase (PI3K), which are upstream of Akt-HTT (*Cohen et al., 2009*; *Humbert et al., 2002*; *Martínez-Mármol et al., 2019*). Prior to this study, the JNK-interacting protein 1 (JIP1) was identified as a scaffold for APP (*Muresan and Muresan, 2005*). JIP1 determines the directionality of APP trafficking through its phosphorylation at a JNK-dependent phosphorylation site (*Fu and Holzbaur, 2013*) and could regulate amyloid-independent mechanisms of AD pathogenesis (*Margevicius et al., 2015*). This study and ours highlight the complexity of the regulation of APP trafficking in neurons, with different kinases (JNK and Akt) responding to specific signaling pathways.

Defects in APP trafficking could also contribute to synaptic defects observed in HD, as we found that the Akt-HTT pathway is down-regulated in HD patient brain samples and lymphoblasts as well as in HD rodent models (*Colin et al., 2005*; *Humbert et al., 2002*). Several studies have reported a notable reduction in the number of synapses particularly within the corticostriatal circuit, which is the most profoundly affected in HD (*Virlogeux et al., 2018*). A better understanding of HTT-APP relationship could help unravel mechanisms of interest for both Huntington's disease and Alzheimer's disease.

## Contact for reagent and resource sharing

Further information and requests for resources and reagents should be directed to and will be fulfilled by the Lead Contact, Frédéric Saudou (Frederic.saudou@inserm.fr).

## Materials and methods

### Mice

The APPswe;PS1L166P (Tg Thy1-*APPKM670/671 NL*;Thy1-*PS1L166P* referred as APPPS1) mouse strain (#21; C57Bl6/J background) was obtained from Dr. M. Jucker's laboratory (*Radde et al., 2006*). Heterozygous $Htt^{S421A/+}$ Knock-In mice (C57Bl6/J background) were generated at the Mouse Clinical Institute (Strasbourg, France) by introduction of a point mutation into Exon 9 (AGC >GCC, Ser >Ala) or (AGC >GAC, Ser >Asp) concomitant with introduction of repeated regions LoxP_Neo_-LoxP in intron nine for genotyping and a FseI restriction site in intron eight for cloning purpose. Homozygous $Htt^{S421A/S421A}$ Knock-In mice ($HTT_{SA}$) were generated and did not show obvious phenotype as shown previously (*Thion et al., 2015*) and in this study. Transgenic APPPS1 mice in a $Htt^{S421A/S421A}$ ($HTT_{SA}$) genetic background mice were obtained by crossing homozygous $Htt^{S421A/S421A}$ mice with transgenic APPPS1 mice thus generating APPPS1;$Htt^{S421A/+}$ mice that were crossed with heterozygous $Htt^{S421A/+}$ mice to obtain APPPS1;$Htt^{S421A/S421A}$ mice, heretofore referred to as APPPS1/$HTT_{SA}$ mice. WT mice used for backcrossing and mouse amplification are C57Bl6/J mice from Charles River Laboratories (L'Arbresle, France). WT mice used for behavioral and biochemical experiments are littermates of APPPS1/$HTT_{SA}$ and, of $HTT_{SA}$ and APPPS1 mice respectively.

The general health of the mice was regularly checked and body weights were assessed weekly throughout the experimental period. Animals were held in accordance with the French Animal Welfare Act and the EU legislation (Council Directive 86/609/EEC) and the ARRIVE (Animal Research: Reporting of In Vivo Experiments) guidelines. The French Ministry of Agriculture and the local ethics committee gave specific authorization (authorization no. 04594.02) to BD to conduct the experiments described in the present study.

To evaluate the effects of $HTT_{SA}$ mutation on a WT background, a total of 23 mice were used. To evaluate the effects of $HTT_{SA}$ mutation on an APPPS1 transgenic background, a total of 84 mice were used. Only male mice were studied, to avoid any potential effects of the estrus cycle on behavioral responses. Behavioral phenotypes were analyzed between 12 and 15 months of age, a time at which behavioral defects are observed in APPPS1 mice (*Radde et al., 2006*).

### Behavioral and cognitive evaluation

The behavioral testing battery consisted of: primary modified SHIRPA screen (*Rogers et al., 1997*), open field, elevated plus maze, novel object recognition, Morris water maze, and grip strength tests.

All tests were performed during light phases of the diurnal cycle. Mice were group housed (4–6/cage) and had free access to food and water except during experiments. They were transported to the behavioral testing room and allowed to acclimate for at least 1 hr prior to initiating experiments.

## Novel object recognition test (NOR)

The object recognition test is based on the natural tendency of rodents to spend more time investigating a novel object than a familiar one (*Ennaceur and Delacour, 1988*). The choice to explore the novel object reflects the use of recognition memory. Our protocol is similar to that previously described (*Scholtzova et al., 2009*). The object recognition test was carried out in an illuminated (30 lux) square gray PVC open field box (50 cm x 50 cm x 30 cm). The test consists of a familiarization session of 15 min in which mice explored the open field arena containing two identical, symmetrically placed objects (A1 and A2). The following day, a training session of 15 min was run with two novel identical objects (B1 and B2). Retention was tested 3 hr after the training session to evaluate object memory. During the retention trial, mice were exposed to a third exemplar of the familiar object (B3) and to a novel object (C1) for 10 min. Behavioral monitoring was done with ANY-maze (Stoelting, USA). The results were expressed as a recognition index, defined as percentage of the time spent exploring the new object over the total time exploring the two objects. Experiments with animals whose exploration was not considered sufficient to allow recognition (less than 6 s of exploration time during training and retention sessions) were discarded from analysis.

## Morris water maze test (MWM)

Spatial learning capacity was tested in the standard hidden-platform Morris water maze (MWM). The maze consisted of a large circular pool (diameter 150 cm) filled with water to a depth of 35 cm. The MWM protocol was adapted from a previous description (*Lo et al., 2013*). Briefly, mice were trained for 11 days to find a hidden platform (10 cm diameter) set at 1 cm beneath the surface of the water at a fixed position in a selected, constant quadrant. The water was opacified with non-toxic white paint (ACUSOL, Brenntag, Belgium) to prevent animals from seeing the platform. The water temperature was maintained at 25–26°C with four thermostatically controlled heaters (Askoll Therm XL 200W, Truffaut, France). The pool was situated at the center of a brightly lit room (~320 lux) with various fixed posters and visual cues placed on the walls to act as distal landmarks. There were four trials per training day with an inter-trial interval of 30 min. The mice were released into water at semi-randomly chosen cardinal compass points (N, E, S, and W). Mice failing to reach the platform within 90 s were gently guided to the platform and were left on it for 15 s, before being dried and returned to their home cages. Two days of rest were given after the 5th and 10th day of training. On the 11th day of training (i.e.: 72 hr after last training session), a probe trial was performed to evaluate robustness of spatial memory. During the probe trial, the platform was removed and mice were released into the pool from the side diagonally opposite to where the platform was located and allowed to swim freely for 90 s.

During all testing phases, a video camera was positioned above the pool for trial recording and the ANY-maze videotracking software was used. Rather than measuring latency or distance traveled, which could be biased by variations in swim speeds and path tortuosity, we analyzed the Corrected Integrated Path Length (CIPL) (*Gallagher et al., 1993*) to assess learning during the 11 training days. During the probe test the percent of time spent in each quadrant was assessed.

## Antibodies, plasmids and lentiviruses

Antibodies used are anti-: HTT (clone D7F7, Cell Signaling; 5656), pHTT-S421 pAb 3517 (*Colin et al., 2008*), GFP (for western blotting, Institut Curie, A-P-R#06), SNAP25 (AbCam, sb24737), Synaptophysin (Cell Signaling; s5768), PSD95 (Millipore; mab1598), p38 (AbCam, ab14692), APP (clone 22C11, Millipore; mab348), Lamin B1 (AbCam; ab133741), KHC (clone SUK4, Covance; MMS-188P), p-150 (BD Transduction Laboratories, 610474), MAP-2 (Millipore; AB5622), GFP (for Immunofluorescence experiments, AbCam; Ab13970), HA (clone 6E2, Cell Signaling, mAb#2367), Tubulin (Sigma; t9026) and GAPDH (Sigma; G9545).

APP-mCherry (*Marquer et al., 2014*) plasmid was cloned into pSIN lentiviral vector (*Drouet et al., 2009*) by Gateway system (Life Technology) using sense primer 5'-GGGGACAAG TTTGTACAAAAAAGCAGGCTTCGAATTCTGCAGTCGACGG-3' and anti-sense primer 5'-

GGGGACCACTTTGTACAAGAAAGCTGGGTCGCGGCCGCCCTACTTGTACA-3' and recombination. We verified whether APP-mCherry is cleaved in neuronal culture by endogenous secretases but did not find aberrant cleavage of APP in our experimental conditions indicating that most of the APP-containing vesicles correspond to full length APP (data not shown). Plasmids coding for pHRIG-Akt1 (Akt-CA) and pHRIG-AktDN (Akt-N) were gifts from Heng Zhao (Addgene plasmids # 53583 and 53597 respectively). MAP2-GFP was previously described (*Liot et al., 2013*). Lentiviruses encoding the first 480 amino acids of HTT with 17Q and with the S421A mutation have been previously described (*Pardo et al., 2006*). Plasmid coding for GFP lentivector was a gift from Dr J. M. Heard. Lentivectors were produced by the ENS Lyon Vectorology Facility with titer higher than $10^8$ UI/ml.

## Vesicular transport imaging into microchambers

Cortical neurons were isolated from mouse embryos (E15.5) according to *Liot et al., 2013*. Neurons were seeded on 12-well plate coated with poly-L-lysine (1 mg/ml) or into microchambers coated with poly-D-lysine (0.1 mg/ml; presynaptic and synaptic compartments) or poly-D-lysine and laminin 10 µg/ml (Sigma; postsynaptic compartment) and cultured at 37°C in a 5% CO2 incubator for 13 days. For dendritic trafficking, mouse neurons were transfected before plating with 5 µg of MAP2-GFP plasmid using a Nucleofector (Lonza) according to the manufacturer's specifications. After 8 days in vitro (DIV8), neurons were transduced as previously described (*Bruyère et al., 2015*) into presynaptic neuron chamber for axonal transport analysis or into the postsynaptic neuron chamber for the dendritic transport analysis. Acquisitions were done at DIV13 on microgrooves, at the limit of the synaptic compartment, at 5 Hz for 1 min on inverted microscope (Axio Observer, Zeiss) coupled to a spinning-disk confocal system (CSU-W1-T3; Yokogawa) connected to an electron-multiplying CCD (charge-coupled device) camera (ProEM+1024, Princeton Instrument) at 37°C and 5% $CO_2$. Vesicle velocity, directional net flux and vesicle number were measured on 100 µm of neurite using KymoTool Box ImageJ plugin as previously described (*Virlogeux et al., 2018*). Vesicle velocity corresponds to segmental anterograde or retrograde velocity. Directional net flux is the anterograde cumulative distance minus the retrograde cumulative distance. Regarding vesicle number, a vesicle is considered anterograde when the distance travelled by one vesicle is more anterograde than retrograde.

## Detection of active synapses

Neurons were seeded into microfluidic devices and transduced at DIV1 with GFP lentivirus. The synaptic chamber was incubated at indicated time with 10 µM of FM4-64 styryl dye (ThermoFischer Scientific) into high KCl Tyrode solution (2 mM NaCl; 50 mM KCl; 2 mM $CaCl_2$; 1 mM $MgCl_2$; 10 mM Glucose and 1 mM Hepes buffer pH7.4) during 1 min at 37°C. After three washes with Tyrode solution (150 mM NaCl; 4 mM KCl; 2 mM $CaCl_2$; 1 mM $MgCl_2$; 10 mM Glucose and 1 mM Hepes buffer pH7.4) containing inhibitors of additional firing (1 mM kynurenic acid and 10 mM $MgCl_2$), acquisitions were made on inverted microscope (Axio Observer, Zeiss) coupled to a spinning-disk confocal system (CSU-W1-T3; Yokogawa) connected to an electron-multiplying CCD (charge-coupled device) camera (ProEM+1024, Princeton Instrument) at 37°C and 5% CO2 with z stacks of 5 µm.

## Quantification of exocytosis rate

COS-1 cells were plated at low density on glass coverslips and transfected with APP-SEP (super ecliptic pHluorin) and with pARIS HTT or pARIS $HTT_{SA}$ (*Pardo et al., 2010*) using calcium phosphate. Acquisitions were made the day after transfection at 5 Hz during 1 min using an inverted microscope (Elipse Ti, Nikon) with a X60 1.42 NA APO TIRF oil-immersion objective (Nikon) coupled to a CCD camera (CoolSnap, Photometrics) and maintained at 37°C and 5% CO2. Analysis was done on area delimited by cell edges and exocytosis rate was quantified using ExocytosisAnalyser macro on ImageJ developed by Marine Scoazec.

## Quantification of APP in the synaptic chamber

Thirteen days after plating, media from synaptic or presynaptic compartments of 9 microchambres per condition were removed. Lysis buffer containing 4 mM Hepes, pH 7.4, 320 mM sucrose and protease inhibitor cocktail (Roche) mixed with 1X Laemmli buffer was added on synaptic and

presynaptic compartments during 30 min. Harvested media containing lysed neurites and synapses were analyzed on western blot.

## Fractionation of Synaptosomes, PSD and non-PSD fractions

Synaptosome purification was performed as previously described (*Frandemiche et al., 2014*). Cortex was homogenized in cold buffer containing 4 mM Hepes, pH 7.4, 320 mM sucrose and protease inhibitor cocktail (Roche). Homogenates were cleared at 1000 g for 10 min to remove nuclei and large debris. The resulting supernatants were concentrated at 12,000 g for 20 min to obtain a crude membrane fraction, which was then resuspended twice (4 mM HEPES, 1 mM EDTA, pH 7.4, 20 min at 12,000 g). Then, the pellet was incubated (20 mM HEPES, 100 mM NaCl, 0.5% Triton X-100, pH 7.2) for 1 hr at 4°C with mild agitation and centrifuged at 12,000 g for 20 min to pellet the synaptosomal membrane fraction. The supernatant was collected as the non-postsynaptic density membrane fraction (non-PSD) or Triton-soluble fraction. The pellet was then solubilized (20 mM HEPES, 0.15 mM NaCl, 1% Triton X-100, 1% deoxycholic acid, 1% SDS, pH 7.5) for 1 hr at 4°C and centrifuged 15 min at 10,000 g. The supernatant contained the PSD or Triton-insoluble fraction. The non-PSD integrity was checked by synaptophysin immunoblotting and the PSD fraction was confirmed by the PSD-95 immunoblotting enriched in this compartment.

## Synapse analysis by electron microscopy

Mice were anesthetized at 19 months of age with pentobarbital (120 mg/kg) and then transcardially perfused with phosphate-buffered saline solution. Hippocampi were dissected and fixed with 2% glutaraldehyde and 2% paraformaldehyde in 0.1 M phosphate buffer pH 7.2 during 48 hr at 4°C; the CA1 area was dissected under the binocular and further fixed during 72 hr in the same solution. Samples were then washed with buffer and post-fixed with 1% Osmium tetroxyde and 0.1 M phosphate buffer pH 7.2 during 1 hr at 4°C. After extensive washing with water, cells were further stained with 1% uranyl acetate pH four in water during 1 hr at 4°C before being dehydrated through graded ethanol (30%–60%-90%-100%-100–100%) and infiltrate with a mix of 1/1 epon/alcohol 100% during 1 hr and several bath of fresh epon (Flukka) during 3 hr. Finally, samples were included in a capsule full of resin that was let to polymerize during 72 hr at 60°C. Ultrathin sections of the samples were cut with an ultramicrotome (Leica), sections were post-stained with 5% uranyl acetate and 0.4% lead citrate before being observed with a transmission electron microscope at 80 kV (JEOL 1200EX). Images were acquired with a digital camera (Veleta, SIS, Olympus) and morphometric analysis was performed with iTEM software (Olympus). Quantification of synaptic density was done on axon-free neuropil regions (*Zhang et al., 2015*).

## Immunostaining into microchambers

Neurons within microchambers were fixed with a PFA/Sucrose solution (4%/4% in PBS) for 20 min at room temperature (RT). The fixation buffer was rinsed three times with PBS and neurons were incubated for 1 hr at RT with a blocking solution (BSA 1%, normal goat serum 2%, Triton X-100 0.1%). For PSD95 and synaptophysin immunofluorescence, the synaptic compartment was then incubated overnight at 4°C with primary antibodies PSD95 (Millipore, #MAB1598, 1:1,000) and Synaptophysin (Abcam, #AB14692, 1:200). For GFP and MAP-2 immunofluorescence, all compartments were incubated with primary antibodies against MAP-2 (Millipore; AB5622) and GFP (AbCam; Ab13970). After washing with PBS, appropriate fluorescent secondary antibodies were incubated for 1 hr at RT. The immunofluorescence was maintained in PBS for a maximum of one week in the dark at 4°C. Immunostainings were acquired with a X63 oil-immersion objective (1.4 NA) using an inverted confocal microscope (LSM 710, Zeiss) coupled to an Airyscan detector to improve signal-to-noise ratio and to increase resolution. Juxtaposition analyses were performed using ImageJ. Airyscan images were thresholded to remove non-specific signal and an area of interest of at least 100 μm in length was defined around neurites. The number of synaptophysin spots overlapping, juxtaposed or separated by no more than two pixels (130 nm) to PSD95 spots were counted manually. Results were expressed as a function of neurite length and were normalized to 100 μm and WT condition. Each condition was tested using at least two chambers per culture from at least two independent cultures. In each chamber, three fields were analyzed in which at least 3 regions of interest were selected (n = number of fields).

## MRI analyses

### Brain preparation for ex vivo MRI acquisitions

Skulls were processed as described in *Pagnamenta et al., 2019*. Briefly, four male mice of each genotype aged between 8 and 11 weeks were transcardially perfused with 4% paraformaldehyde solution in phosphate buffered saline containing 6.25 mm of Gd-DOTA (Guerbet Laboratories, Roissy, France). This contrast agent is added to reduce the MRI acquisition time. Skin and head muscles were removed to expose the skull, which was then immersed in the fixing solution for 4 days. The skull was then transferred to a Fomblin (FenS chemicals, Goes, Netherlands) bath for at least 7 days for the distribution of Gd-DOTA to be homogeneous throughout the whole brain.

### MRI acquisitions

Ex vivo 3D MRI acquisitions were performed as described in *Pagnamenta et al., 2019*. Briefly, skulls were put in a 9,4 T MRI (Bruker Biospec Avance III; IRMaGe facility) and a volume coil for transmission and a head surface cryocoil for reception were used.

To quantify brain volume, the brain was segmented with a 3D $T_{1W}$ gradient-echo MRI sequence (repetition time: 35.2 ms, echo time: 8.5 ms, flip angle: 20 degrees, field of view:12 $\times$ 9$\times$18.1 mm3, isotropic spatial resolution: 50 µm, four signal accumulations, total acquisition time per brain: 2 hr 32 min).

### Quantitative analysis of brain volumes

Quantitative analysis of brain volume was performed as described in *Pagnamenta et al., 2019*. Briefly, from 3D $T_{1W}$ gradient-echo MRI images, brain structures were delimited manually with the help of Allen mouse brain atlas (http://atlas.brain-map.org/atlas) every five slices by defining regions of interest (ROIs) on the coronal orientation using Fiji software. Then, interpolation was applied using the segmentation editor plug-in (http://fiji.sc/Segmentation_Editor) for the brain structures (total brain, cortex, hippocampus and striatum) to be reconstructed. A color was attributed to each structure and its 3D reconstruction and volume were determined using the Voxel counter plug-in in the Fiji software (https://imagej.net/3D_Viewer). Volume was calculated as following: number of voxels x voxel volume. All segmentations were done blind to the genotype. Mann and Whitney test was used for comparison.

## Statistical analyses

All cellular biology experiments were repeated in at least in three different batches of cultures. Normality of data distribution was verified by graphical analysis of the data distribution and residues. For data with assumed normal distribution, groups were compared by parametric tests followed by post hoc analyses for multiple comparisons. Non-parametric tests were used for western blot analyses and Aβ dosages. *p<0.05; **p<0.01; ***p<0.001; ****p<0.0001; ns, not significant. Statistical calculations were performed using GraphPad Prism 6.0.

## Acknowledgements

We thank Luc Buée, Alain Buisson, Bassem Hassan, Subhojit Roy, Rémy Sadoul, and members of the Saudou, Humbert and Potier labs for comments; Vicky Brandt for critical reading; Caroline Benstaali, Anne Bertrand, Camille Brochier, Aurélie Genoux, Félicie Lorenc, Jessica Roland, Chiara Scaramuzzino, Marine Scoazec and Gisela Zalcmann for technical help and/or initial experiments; Daniel Choquet for the gift of Super-Ecliptic-pHluorin-APP plasmid, Y Saoudi and the GIN imaging facility (PIC-GIN) for help with image acquisitions; G Froment, D Nègre, and C Costa from SFR Biosciences (UMS3444/CNRS, US8/Inserm, ENS de Lyon, UCBL) for lentivirus production; Emmanuel Barbier and Olivier Montigon (UMS 3552/US17 IRMage) for MRI acquisitions; the technical staff of the PHENO-PARC platform and the HISTOMICS platform of the ICM for behavioral and histological studies. This work was supported by grants from Agence Nationale de la Recherche: ANR-12-MALZ-0004 HuntA-beta, FS and MCP; ANR-14-CE35-0027-01 PASSAGE, FS; ANR-15-IDEX-02 NeuroCoG (FS) and ANR-10-IAIHU-06 (MCP) in the framework of the 'Investissements d'avenir' program), Fondation pour la Recherche Médicale (FRM, DEI20151234418, FS; DEQ20170336752, SH), Fondation pour la Recherche sur le Cerveau (FRC)(FS), Fondation Bettencourt Schueller (FS) and AGEMED program

from INSERM (FS and SH). IRMaGe is partly funded by 'Investissements d'Avenir' run by the French National Research Agency, grant 'Infrastructure d'avenir en Biologie Santé' (ANR-11-INBS-0006). FS laboratory is member of the Grenoble Center of Excellence in Neurodegeneration (GREEN). HV was supported by a PhD fellowship from Association Huntington France and by a FRM fellowship (FDT201904008035)

## Additional information

### Funding

| Funder | Grant reference number | Author |
| --- | --- | --- |
| Agence Nationale de la Recherche | ANR-12-MALZ-0004 HuntAbeta | Marie-Claude Potier Frédéric Saudou |
| Fondation pour la Recherche Médicale | DEI20151234418 | Frédéric Saudou |
| Fondation pour la Recherche sur le Cerveau | | Frédéric Saudou |
| Inserm | AGEMED | Sandrine Humbert Frédéric Saudou |
| Fondation Bettencourt Schueller | | Frédéric Saudou |
| Association Huntington France | | Hélène Vitet |
| Agence Nationale de la Recherche | ANR-14-CE35-0027-01 PASSAGE | Frédéric Saudou |
| Agence Nationale de la Recherche | ANR-15-IDEX-02 NeuroCoG | Frédéric Saudou |
| Agence Nationale de la Recherche | ANR-10-IAIHU-06 | Marie-Claude Potier |
| Fondation pour la Recherche Médicale | DEQ20170336752 | Sandrine Humbert |
| Fondation pour la Recherche Médicale | FDT201904008035 | Hélène Vitet |

The funders had no role in study design, data collection and interpretation, or the decision to submit the work for publication.

### Author contributions

Julie Bruyère, Conceptualization, Formal analysis, Investigation, Visualization, Methodology, Writing - original draft, Writing - review and editing; Yah-Se Abada, Formal analysis, Investigation, Visualization, Methodology, Writing - original draft, Writing - review and editing; Hélène Vitet, Formal analysis, Investigation, Visualization, Methodology, Writing - review and editing; Gaëlle Fontaine, Aurélia Cès, Investigation, Acquisition of data, or analysis and interpretation of data, Final approval of the version to be published; Jean-Christophe Deloulme, Formal analysis, Investigation, Visualization, Methodology; Eric Denarier, Formal analysis, Methodology; Karin Pernet-Gallay, Formal analysis, Investigation, Visualization; Annie Andrieux, Supervision, Writing - review and editing; Sandrine Humbert, Conceptualization, Supervision, Methodology, Writing - review and editing; Marie-Claude Potier, Conceptualization, Supervision, Funding acquisition, Methodology, Writing - original draft, Writing - review and editing; Benoît Delatour, Conceptualization, Supervision, Visualization, Methodology, Writing - original draft, Writing - review and editing; Frédéric Saudou, Conceptualization, Supervision, Funding acquisition, Visualization, Methodology, Writing - original draft, Project administration, Writing - review and editing

### Author ORCIDs

Hélène Vitet https://orcid.org/0000-0003-2899-1064
Jean-Christophe Deloulme http://orcid.org/0000-0002-2234-5865

Eric Denarier [ID] http://orcid.org/0000-0002-4169-397X
Annie Andrieux [ID] http://orcid.org/0000-0002-4022-6405
Sandrine Humbert [ID] https://orcid.org/0000-0002-9501-2658
Frédéric Saudou [ID] https://orcid.org/0000-0001-6107-1046

## Ethics

Animal experimentation: Animals were held in accordance with the French Animal Welfare Act and the EU legislation (Council Directive 86/609/EEC) and the ARRIVE (Animal Research: Reporting of In Vivo Experiments) guidelines. The French Ministry of Agriculture and the local ethics committee gave specific authorization (authorization no. 04594.02) to BD to conduct the experiments described in the present study.

## Decision letter and Author response

Decision letter https://doi.org/10.7554/eLife.56371.sa1
Author response https://doi.org/10.7554/eLife.56371.sa2

## Additional files

### Supplementary files

• Supplementary file 1. The modified SHIRPA primary screen in WT and HTT$_{SA}$ mice. Results are presented in percentages unless otherwise indicated. No significant differences between genotypes were observed.

• Transparent reporting form

### Data availability

All data generated or analysed during this study are included in the manuscript and supporting files.

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

# Appendix 1

## Supplemental Methods

### Vesicular fractionation

Three 6/7-month-old mice were used per genotype. The protocol was adapted from *Zala et al., 2013*. Briefly, frozen brains were homogenized in 700 uL of homogenization buffer (320 mM sucrose) by triturating with a Dounce homogenizer. Then, samples were centrifuged twice at 47.000 rcf for 10 min and the two supernatants were combined (S1). From S1, another ultracentrifugation was performed at 120.000 rcf for 40 min to obtain S2 and P2. The S2 fraction was put in another tube and 280 µL of 700 mM sucrose buffer was added under the S2 fraction. After a 2 hr centrifugation at 260 000 rcf, S3 and P3 were obtained. P3 fraction was resuspended in 50 µL of Re-suspension buffer (10 mM HEPES pH 7.3 and 320 mM sucrose). Antibodies used for western blot analysis are KHC (clone SUK4, Covance; MMS-188P,) and p150 (BD Transduction Laboratories, 610474).

### Primary Screen (modified SHIRPA)

The testing was carried out using a modified version of the standard protocol (*Rogers et al., 1997*). HTT$_{SA}$ and WT littermate control mice were examined at 12 months of age. The primary screen began by observing undisturbed behavior in a viewing jar (clear Perspex cylinder, 15 cm x 11 cm) for 30 s (section 1). Thereafter, the mouse was transferred in the arena (55 cm x 33 cm x 18 cm) for testing of transfer arousal and observation of normal behavior (section 2). The observer also looked for any manifestation of bizarre or stereotyped behavior, convulsions and indications of spatial disorientation. This was followed by a sequence of manipulations using tail suspension and the grid across the width of the arena (section 3). To complete the assessment, the animal was restrained in a supine position to record autonomic behaviors (section 4).

### Open field activity (OF)

The open field test was used to assess locomotor activity (and anxiety-related behaviors). Mice were tested in a homogeneously illuminated (50 lux) circular open field arena made of white plastic (diameter: 54 cm) with 30 cm-high walls. Monitoring was done by an automated video tracking system (AnyMaze, Stoelting, Wood Dale, IL, USA). The main behavioral parameters analyzed during a single 10 min session in the OF were the total traveled distance and also the center-to-periphery exploration ratio.

### Grip force test

Mice were scruffed by the lower back and lowered towards a mesh grip piece attached to a force gauge (Bioseb) until the animal grabbed it with both front paws. The animal was then lowered toward the platform and gently pulled straight back with consistent force until it released its grip. The forelimb grip force was recorded on the strain gauge.

### Elevated plus maze (EPM)

The EPM was made of beige PVC and the center of the field illuminated at 70 lux. The apparatus was elevated 50 cm above floor level and consisted of four arms (35 cm ×5 cm). Two of the arms contained 15 cm-high walls (enclosed arms) and the other two had no walls (open arms). Each mouse was placed in the middle section facing an open arm and left to explore the maze for a single 5 min session with the experimenter out of view. Animals were video-recorded and their behavior automatically analyzed with the ANY-maze software.

Percent time spent in open arms which is supposed to be inversely correlated to anxiety levels was measured for each mouse.

## Neuropathology

Following completion of behavioral testing, at 19 months of age, mice were anesthetized with pentobarbital (120 mg/kg) and then transcardially perfused with phosphate-buffer saline solution. The brain was extracted and carefully weighed on a precision balance. One hemisphere was snap frozen in liquid nitrogen and then stored at −80°C for subsequent biochemical analysis. The other half brain was fixed by immersion in freshly-made formaldehyde solution (3–4 days), then cryoprotected in a 2% - DMSO – 20% glycerol solution and finally cut on a freezing microtome (serial sections of 40 µm of the entire brain).

Amyloid deposits were labeled by standard Congo red staining (30 min in an 80% ethanol solution saturated with Congo red and sodium chloride). Microscopic scans of whole sections (pixel size 0.25 µm$^2$) were acquired with a NanoZoomer 2.0-RS slide scanner (Hamamatsu Photonics, Hamamatsu, Japan). Amyloid loads were quantified using computer-based segmentation methods using the spot detector plugin of the ICY software (http://icy.bioimageanalysis.org) that automatically calculates the proportion of stained tissue (p=stained area/total area), providing unbiased stereological measurements.

## Immunolabelling

For immunostaining, free-floating sections were washed in PBS 0.1M to remove cryoprotectant. The sections were treated with hydrogen peroxide for 10 min to quench endogenous peroxidase activity, permeabilized with 0.25% Triton X-100 in PBS 0.1M (PBSTx) for 20 min, pre-incubated in a 5% PBS-Tx normal goat serum (NGS) blocking solution and then incubated overnight at room temperature (RT) with the following primary antibodies : a biotinylated mouse anti-Aβ (4G8) (1:3000, Covance Antibody Products), a rabbit polyclonal antibody recognizing amyloid fibrils and fibrillary Aβ oligomers (**OC**) (1:3000, StressMarq Biosciences) and a rabbit polyclonal anti-prefibrillar Aβ oligomers (A11) (1:1000, generous gift of Dr. Kayed Rakez). The sections were incubated with a secondary biotinylated goat anti-rabbit antibody at RT for 90 min (this step was omitted for 4G8 antibody that was already biotinylated). Tissues were then incubated in the Vector Elite avidin-biotin peroxidase kit (1:800) for 90 min at RT. Finally, after washes in PBS-Tx and Tris 0.1M solutions, immunoreactivity was revealed using diaminobenzidine (DAB) as chromogen to visualize the reaction product. The sections were then mounted on Superfrost slides, dehydrated in a series of alcohols (30%, 50%, 70%, 2 × 90% and 2 × 100%), cleared in xylene, and coverslipped with EUKITT mounting medium.

## Analysis

Microscopic scans of immunostained brain sections were acquired with a **NanoZoomer 2.0-RS slide scanner** (Hamamatsu Photonics) at 40X magnification (pixel size 0.25 µm$^2$). Selected regions of interest (ROIs) were delineated by using the Paxinos and Franklin Mouse Brain Atlas: sensori-motor (SM), Frontal (FR) and the hippocampus (HPC). ROIs were assessed across 2 to 6 consecutive serial sections (depending on structure) and were manually outlined on digitized sections. Computer-based segmentation methods were applied for 4G8 and OC immunostaining using the Best threshold plugins of the **ICY**) software (https://icy.bioimageanalysis.org) that automatically calculate the proportion of stained tissue (p=stained area/total area) in each ROIs. For A11 immunostained sections, Aβ loads were calculated using the **Ilastik** interactive learning and segmentation toolkit software (https://ilastick.org/index.html). The frontal cortex, the sensori-motor cortex and the hippocampus brain regions were manually outlined on digitized sections. To evaluate the 4G8-detected Aβ loads, reference background staining of the corpus callosum was used to binarize the digitized image to 8-bits black and white image. The mean number of thresholded pixels per ROI was automatically

calculated using an ICY image analysis software script. A minimum of 3 sections per brain ROIs per animal were analyzed and counting's were reported to the overall ROI surface to provide the 4G8 amyloid loads.

### Aβ42 dosages

Hemi-forebrains (~200 mg) were harvested in 500 µl of solution containing 50 mM Tris-HCl (pH 7.6), 0.01% NP-40, 150 mM NaCl, 2 mM EDTA, 0.1% SDS, 1 mM phenylmethylsulfonyl fluoride (PMSF), and protease inhibitor cocktail (Sigma). Soluble, extracellular-enriched proteins were collected from mechanically homogenized lysates following centrifugation for 10 min at 3,000 g. Cytoplasmic proteins were extracted from cell pellets mechanically dissociated with a micropipettor in 500 µl buffer containing 50 mM Tris-HCl (pH 7.6), 150 mM NaCl, 0.1% Triton X-100 following centrifugation 90 min at 11,000 g. Supernatant was collected for dosages. Supernatants from cortical neurons plated in microchambers were collected on ice in polypropylene tubes (Corning, Corning, NY, USA) containing a protease inhibitor cocktail (Roche) and were then stored at −80℃ Concentration of Aβ peptides were measured by Electro-Chemiluminescence Immuno-Assay (ECLIA) performed according to the manufacturer's instructions Meso Scale Discovery (MSD). Briefly, samples were analyzed using MSD SECTOR Imager 2400 (Meso Scale Discovery, Gaithersburg, MD, USA), with the Rodent Aβ triplex kit (from MSD) on carbon 96-well plates. 100 µl of blocking buffer solution were added to avoid non-specific binding. The plates were then sealed, wrapped in tin foil, and incubated at room temperature on a plate shaker (300 rpm) for 1 hr. Wells were then washed three times with washing buffer, and 25 µl of the standards and samples were then added to the wells, followed by an Aβ-detecting antibody at 1 µg/ml (MSD) labelled with a Ruthenium (II) trisbipyridine N-hydroxysuccinimide ester; this detection antibody was 4G8. Plates were then aspirated and washed three times. MSD read buffer (containing TPA) was added to wells before reading on the Sector Imager. A small electric current passed through a microelectrode present in each well producing a redox reaction of the Ru2+ cation, emitting 620 nm red light. The concentration of Aβ was calculated for each sample, using dose–response curves, the blank being cell-less culture medium. All the conditions were tested in duplicate. Aβ levels were normalized with total amount of proteins quantified by Bradford dosage.

