## [Decision Letter]

**Acceptance summary:**

Using S421D Htt and S421A knockin mice, this study nicely demonstrates the functional importance of Htt 421 phosphorylation on axonal transport of amyloid precursor protein. In addition, it demonstrates a role of neuronal cell-autonomous mechanisms, independent of the plaque/microglia-related toxicities, in the development of synaptic and behavioral impairments in amyloid AD models.

**Decision letter after peer review:**

[Editors’ note: the authors submitted for reconsideration following the decision after peer review. What follows is the decision letter after the first round of review.]

Thank you for choosing to send your work, "Axonal Akt-Huntingtin pathway determines synaptic Aβ levels", for consideration at *eLife*. Your article has been reviewed by three peer reviewers, one of whom is a member of our Board of Reviewing Editors, and the evaluation has been overseen by a Senior Editor. Although the work is of interest, we regret to inform you that the findings at this stage are too preliminary for further consideration at *eLife*.

All reviewers found that your work addresses an important question – HTT's role in axonal transport, and using a physiologically relevant model system. However, there are key areas where it was felt that the status of the current work fell short. Thus, given that the amount of work needed to improve this study will take more than two months, we regret that we must reject the current manuscript. Since all reviewers noted the potential of this work, however, we would be happy to consider a new submission in the future as long as it addresses the key issues. If you select to submit a new and revised manuscript, it is critical that the Htt S421A knockin mice, a novel model, be characterized in more depth; e.g. are there developmental defects that might explain synaptic changes rather than the alterations in trafficking. In addition, since previous work linked Htt to trafficking of APP the conceptual advance of the current work is limited. To enhance the conclusion that this Htt/S421 phosphorylation pathway is relevant to AD direct data linking this pathway to AD needs to be presented, perhaps by crossing the Htt S421A mice to an amyloid AD mouse model.

Please note that we aim to publish articles with a single round of revision that would typically be accomplished within two months. This means that work that has potential, but in our judgment would need extensive additional work, will not be considered for in-depth review. We do not intend any criticism of the quality of the data or the rigor of the science. We wish you good luck with your work and we hope you will consider *eLife* for future submissions.

Reviewer #1:

This utilizes an elegant microfluidic system to examine the role of the Htt protein in regulating axonal versus dendritic transport of APP. Previous support the concept that Htt regulates APP transport via phosphorylation of Htt at ser 412, but the specifics of its role and its mechanistic aspects remain unclear. Strengths of this study include the use of full length Htt in an experimental model that closely replicates physiological conditions of axonal and dendritic transport of APP. The basic finding of this study is that ser 421 phosphorylation promotes APP anterograde transport in axons but not in dendrites. In addition, they present data supporting the idea that Akt, a kinase previously shown to phosphorylate ser421, regulates anterograde transport of APP in axons. Unfortunately, to this reviewer's perspective, the potentially most important data are the least impressive. In Figures 4-6 data are presented on the ability of this ser421-Htt pathway to impact APP targeting to the synapse and numbers of synapses. While the impact is statistically significant, effect size blocking ser421 phosphorylation has on AB transport and synaptic number is very subtle. This leaves this reviewer unsure as to the biological relevance of this pathway on regulating synaptic aspects in vivo.

Reviewer #2:

The manuscript by Bruyere and colleagues investigated the effect of Huntingtin phosphorylation at Ser421 on axonal APP transport and synaptic Aβ levels in vitro and in vivo. With chamber separated, primary neuronal cultures, the authors successfully measured the dynamics of APP trafficking in cortical axons and dendrites from synaptically connected cultures. They then used this system to demonstrate the absence of phosphorylation on Ser421 of Htt increased retrograde transport of APP in the axons, while its dendritic transport was unaffected. The authors furthered their findings with Akt inactive and constitutively active transgenes to demonstrate that activated Akt facilitates APP anterograde transport and this phenotype is dependent on the phosphorylation of Htt-Ser421. The authors used a previously described Htt S421A knockin mouse model (HttSA) that allows for the study of this phosphorylation site in the context of the intact Htt gene and full-length protein as a source for their primary cultures. Using the same co-culture system, they suggest that synaptic APP exocytosis is reduced in HttSA cultures. Consistent with this, reduced amounts of synaptic Aβ in the presynaptic compartment of adult HttSA cortices in vivo was identified. Then by testing combinations of wildtype and HttSA cells cultured in separate but synaptically connected chambers, they found that HttSA results in decreased APP in the synaptic compartment only when cultured on the presynaptic side. Finally, the authors examined the Aβ levels in the synaptosomal fractions of HttSA versus wildtype cortex and measured changes in synaptic size and density in CA1, finding a reduction in Aβ and smaller but increased density of synapses in CA1. Overall, the authors concluded that phosphorylation of Htt at Ser421 by Akt regulates APP trafficking in axons but not dendrites and results in altered synapse size and number.

The question of the physiological function of Htt is an important one for both basic biological and neurodegenerative disease reasons. Similarly, APP trafficking is tied to both basic and neurodegenerative disease processes and tying these proteins together mechanistically would be a significant advance. The strength of the manuscript lies in the use of the chambered primary culture system to physically isolate the presynaptic cells, synaptic compartment, and postsynaptic cells and combine this with knockin HttSA mouse cells and transfection/infection of wildtype cells to convincingly demonstrate altered APP trafficking and exocytosis in presynaptic HttSA cells. In addition, the use of the HttSA mouse versus wildtype to characterized changes in Aβ and synaptic APP is informative. However, there are a few major issues need to be addressed to strengthen the study.

1) Major conclusions in this study rely on the precise compartmentalization of axons and dendrites in the microfluidic device. The evidence for the "presynaptic chamber" contains only axons and the postsynaptic compartment contains only dendrites is scarce. If these processes are not neatly separated, how can they be certain the kymographs are measuring APP vesicular transport in different neuronal processes?

2) The CA-Akt construct is somewhat artificial. Can they overexpress or knockdown Akt to change APP anterograde trafficking in axons? What happens to manipulate Akt activity or levels to the dendritic trafficking of APP?

3) Was a Htt S421D mouse knockin model generated in parallel to the HttSA model? If so, this mouse could be utilized in primary cortical neuronal culture to determine if APP transport and exocytosis was accelerated and examine if the synaptic changes observed in HttS421A are the opposite in Htt S421D in vivo?

4) Another general concern is whether S421 mutation could change the levels of Htt expression? Can the knockdown or knockout of endogenous Htt mimic the effects of SA mutation in axonal transport?

5) The discussion on the link of Akt-HTT pathway to AD pathophysiology appears tenuous at the best. Synaptic abeta pathology is largely dependent on synaptic activity and the abnormal processing of abeta, but not merely on the levels of APP in the axonal terminals. Moreover, this is a hypothesis that can be readily tested by crossing the HttSA mice with the APP transgenic mice. Without such study, they should avoid speculating the role of Akt-HTT pathway in AD pathophysiology, and limit their interpretation on the pathway in the physiological trafficking of wildtype APP.

Reviewer #3:

In this manuscript, the authors reconstruct a corticocortical neuronal network using a microfluidic device and primary cortical neurons from HTT S421A knock-in mice (HTT S421A/S421A) to investigate the role of HTT S421 phosphorylation in APP transport. First, the authors show that Huntingtin (HTT) phosphorylation at S421 regulates APP transport in axons, but not in dendrites. Second, as the authors previously published that Akt is the kinase that phosphorylates HTT at S421 (Humbert et al., 2002), they validate their previous findings by demonstrating that constitutively active form of Akt (Akt-CA) increases APP transport in the wildtype cortical neurons, but not in HTT S421A neurons, suggesting that the Akt-HTT pathway regulates APP transport in axons. Third, the authors found that APP levels in the presynaptic compartment are decreased in HTTS421A neurons compared to the wildtype. Fourth, they further measure Aβ levels in synaptosomes in HTT S421A neurons and found decreased, and suggest that a decrease in APP and Aβ levels in HTT S421A neurons is due to APP trafficking defects. Lastly, using electron microscopy, the authors examine axonal terminals, spines and synapses in HTT S421A/S421A mouse hippocampus and found shorter postsynaptic density (PSD) length and higher number of synapses compared to the wildtype mice.

This study presents exciting data on the role of HTT phosphorylation in APP transport, but there are some major comments:

Last Figure 6C raises some question on its relevance to APP transport and even more questions on the HTTS421A/S421A knockin mouse phenotype. Are there any differences in cortical neuronal morphology between HTTS421A/S421A knockin mouse verses the wildtype? E.g. in soma size, dendrite length, axonal length, etc. How about overall hippocampus size in HTTS421A/S421A knockin mouse verses the wildtype?

Interestingly, I could not find much phenotypic information about HTT S421A/S421A knockin mice from the previous paper that first generated (Thion . et al., 2015), which raises even more questions about the mice as it is not well characterized.

In addition, it may be important to define the mechanism of APP transport in the axons by further validating previous findings (Colin et al., 2008) such as whether HTT S421 phosphorylation is required for Kinesin-1 recruitment to microtubules and vesicles using HTTS421A/S421A knockin mouse.

Furthermore, I am not sure why the authors focused on APP while it is well known that HTT S421 phosphorylation plays a role in transporting vesicles in general, such as BDNF-containing vesicles in addition to APP containing vesicles (Colin et al., 2008).

---

## [Author Response]

[Editors’ note: the authors resubmitted a revised version of the paper for consideration. What follows is the authors’ response to the first round of review.]

Reviewer #1:This utilizes an elegant microfluidic system to examine the role of the Htt protein in regulating axonal versus dendritic transport of APP. Previous support the concept that Htt regulates APP transport via phosphorylation of Htt at ser 412, but the specifics of its role and its mechanistic aspects remain unclear. Strengths of this study include the use of full length Htt in an experimental model that closely replicates physiological conditions of axonal and dendritic transport of APP. The basic finding of this study is that ser 421 phosphorylation promotes APP anterograde transport in axons but not in dendrites. In addition, they present data supporting the idea that Akt, a kinase previously shown to phosphorylate ser421, regulates anterograde transport of APP in axons. Unfortunately, to this reviewer's perspective, the potentially most important data are the least impressive. In Figures 4-6 data are presented on the ability of this ser421-Htt pathway to impact APP targeting to the synapse and numbers of synapses. While the impact is statistically significant, effect size blocking ser421 phosphorylation has on AB transport and synaptic number is very subtle. This leaves this reviewer unsure as to the biological relevance of this pathway on regulating synaptic aspects in vivo.

All three reviewers felt that we needed to bolster the physiological relevance of the work. We have now conducted a number of new experiments and present the new data throughout the manuscript. In brief, we found that presynaptic APP levels rather than Ab loads correlate with synapse number both in vitro and in vivo (new Figures 5, 6, 7 and 8, Figure 8—figure supplement 1). Crossing HTTSA mice with APP-PS1 mice rescued the latter’s synapse number and improved learning and memory, indicating a link between APP anterograde axonal transport, synapse number and morphology as well as cognitive behaviors in vivo (new Figures 8 and 9).

Reviewer #2:[…] There are a few major issues need to be addressed to strengthen the study.1) Major conclusions in this study rely on the precise compartmentalization of axons and dendrites in the microfluidic device. The evidence for the "presynaptic chamber" contains only axons and the postsynaptic compartment contains only dendrites is scarce. If these processes are not neatly separated, how can they be certain the kymographs are measuring APP vesicular transport in different neuronal processes?

This is an important point, and one we are happy to clarify. Given that dendrites cannot grow over 300µm, and that we capture the transport of vesicles in the distal part (the last 100µm of these microchannels), we are assessing the transport of vesicles in axons only. The fact that axons but not dendrites are present in the distal part is attested by the negative staining for MAP2 (Virlogeux et al., 2018). Conversely, to be certain that we are recording transport of vesicles in dendrites, we are capturing events in the dendritic channels and use MAP2-GFP infection of the postsynaptic neurons to make sure that transport is recorded in dendrites only. The new Figure 1B shows that synapses formed within the synaptic chamber are made from the presynaptic bouton coming from the presynaptic chamber (GFP-transduced neurons) and the postsynaptic dendrite coming from the postsynaptic chamber (MAP2 staining). We conclude that we can measure specifically APP vesicular transport in axons versus dendrites.

We should note that the capacity of microfluidic devices to compartmentalize axons and dendrites is now well established (e.g., (Lehigh, et al., 2017)), and we have carefully characterized the three-compartment device we use here (Virlogeux et al., 2018).

2) The CA-Akt construct is somewhat artificial. Can they overexpress or knockdown Akt to change APP anterograde trafficking in axons? What happens to manipulate Akt activity or levels to the dendritic trafficking of APP?

To answer the second question first: We previously demonstrated that overexpression of Akt or treatment with the Akt physiological activator IGF-1 enhances anterograde transport (Colin et al., 2008, Zala et al., 2008). Here, we used the Akt-CA (constitutively active) construct, which significantly enhances Akt activity in neurons, because – though we agree it is somewhat artificial – it strongly activates Akt and it reliably induces HTT phosphorylation (unlike, say, using IGF1). This enables us not only to avoid the activation of other IGF-1-dependent pathways but also to demonstrate that since the Akt-mediated effect is lost when HTT cannot be phosphorylated, this effect is due solely to Akt-induced HTT phosphorylation.

To make sure that expression of Akt-CA did not produce artifacts, we used an Akt construct that contains a mutation at K179 that blocks its catalytic activity and shows some dominant-negative activity over endogenous Akt. The previous Figure 3A showed a significant difference in anterograde transport of APP in axons between WT + AKT-N versus WT neuron + Akt-CA.

To assess the effect of reducing Akt activity, we measured anterograde transport in WT neurons transduced with GFP, AKT-N and Akt-CA. Akt-N led to a reduction of APP anterograde flux when compared to Akt-CA and GFP, indicating that reducing Akt activity decreases anterograde axonal transport of APP by increasing the speed of retrograde moving vesicles (new Figure 4C). Finally, we show that HTT phosphorylation has no effect on dendrites.

3) Was a Htt S421D mouse knockin model generated in parallel to the HttSA model? If so, this mouse could be utilized in primary cortical neuronal culture to determine if APP transport and exocytosis was accelerated and examine if the synaptic changes observed in HttS421A are the opposite in Htt S421D in vivo?

We did indeed generate a HTT-S421D knock-in mouse. We utilized this model to prepare cortical neuronal cultures and analyze transport in axons and dendrites in parallel to WT and HTTSA neurons. These results (new Figure 3A and B) show that in axons, APP transport in HTT_SD_ neurons is significantly different from the transport in HTT_SA_ neurons for anterograde and retrograde velocities, cumulative distance and net directional flux. Compared to WT, anterograde transport is slightly increased in HTT_SD_ neurons, because the WT neurons are cultured in high percentage of serum and HTT is highly phosphorylated in these conditions (our personal observations). See also the western blot of the new Figure 4A showing that levels of phospho-HTT are similar between GFP and Akt-CA transduced neurons.

We also measured exocytosis by expressing HTT-S421D in Cos cells. The level of exocytosis of APP-SEP in cells transfected with pARIS-HTT_SD_ was similar to that of pARIS-HTT_WT_ cells but significantly different from cells transfected with pARIS-HTT_SA_.

Thus, WT and HTT_SD_ neurons or constructs behave similarly, because HTT is mostly phosphorylated in our experimental conditions. Therefore, in most of the experiments, we compared WT and HTT_SA_ conditions.

**Author response image 1. sa2fig1:** Extended version of Figure 5A showing the comparison in APP exocytosis in cells expressing WT full length HTT (pARIS HTT WT), or version with S421A (pARIS HTT SA) or S421D (pARIS HTT SD) mutations.

4) Another general concern is whether S421 mutation could change the levels of Htt expression? Can the knockdown or knockout of endogenous Htt mimic the effects of SA mutation in axonal transport?

This is another important point, which we clarify in the manuscript: neither the HTT-S421A mutation nor the HTTS421D mutation affects levels of HTT (Ehinger et al., 2020). As previously shown by others and us, silencing HTT reduces both anterograde and retrograde axonal transport (Colin et al., 2008, Gauthier et al., 2004, Her and Goldstein, 2008) while S421 phosphorylation affects directionality ((Colin et al., 2008) and this study). Therefore, knockdown or knockout of endogenous HTT cannot mimic the SA mutation.

5) The discussion on the link of Akt-HTT pathway to AD pathophysiology appears tenuous at the best. Synaptic abeta pathology is largely dependent on synaptic activity and the abnormal processing of abeta, but not merely on the levels of APP in the axonal terminals. Moreover, this is a hypothesis that can be readily tested by crossing the HttSA mice with the APP transgenic mice. Without such study, they should avoid speculating the role of Akt-HTT pathway in AD pathophysiology, and limit their interpretation on the pathway in the physiological trafficking of wildtype APP.

To evaluate the pathophysiological relevance of our findings, we crossed HTT_SA_ mice with APPPS1 mice, which bear mutations that reproduce features of familial Alzheimer’s disease (amyloid plaques, loss of synapses, and cognitive defects (Bittner et al., 2012, Radde et al., 2006, Zou et al., 2015)). We found that blocking HTT phosphorylation in APPPS1/HTT_SA_ mice leads to a reduction of APP levels in the presynaptic compartment, along with complete rescue in synapse number and in post-synaptic density length – all without any effect on Ab accumulation or plaques. Importantly, we found that HTTSA mutation significantly improved spatial and non-spatial memory in APPPS1 mice. We conclude that blocking Akt phosphorylation at HTT S421 reduces APP presynaptic levels and improves learning and memory in APPPS1 mice. These experiments are shown in new Figures 8 and 9 and new Figure 9—figure supplement 1.

Reviewer #3:[…] Last Figure 6C raises some question on its relevance to APP transport and even more questions on the HTTS421A/S421A knockin mouse phenotype. Are there any differences in cortical neuronal morphology between HTTS421A/S421A knockin mouse verses the wildtype? E.g. in soma size, dendrite length, axonal length, etc. How about overall hippocampus size in HTTS421A/S421A knockin mouse verses the wildtype?Interestingly, I could not find much phenotypic information about HTT S421A/S421A knockin mice from the previous paper that first generated (Thion et al., 2015), which raises even more questions about the mice as it is not well characterized.

We now provide a full behavioral characterization of the HTT_SA_ mice. As shown in new Figure 6—figure supplement 1, and in Supplementary file 1, we did not find any motor or cognitive defects in these mice. Also, these mice do not show changes in HTT expression (Ehinger et al., 2020).

There were, however, differences from WT revealed by anatomical MRI (new Figure 6A and B). HTT_SA_ mice showed a significant increase of the whole brain volume (4,8 %) affecting the hippocampus (8,5%) and the cortex (3,7%) but not the striatum. An in-depth characterization of soma size, dendrite length, axonal length, etc. was beyond the scope of this study, but the results would not alter the findings that HTT dephosphorylation affects axonal APP transport, synapse number and morphology.

We believe that this question of reviewer 3 is related to a point raised by the Editors: “Are there developmental defects that might explain synaptic changes rather than the alterations in trafficking?” This is an important but difficult question to answer, since there is evidence that axonal transport affects axonal development, and a dispositive experiment would require selective HTT dephosphorylation in adult mice. We tried to investigate the effect of APP and HTT dephosphorylation in mature circuits in vivo in order to bypass any developmental contribution to the observed effect in adult mice. We stereotaxically injected lentiviruses expressing APP in the entorhinal cortex of 4-month-old animals and counted synapses 2 months later. Unfortunately, we could not detect any changes in the synapse number (data not shown) as compared to what is observed in APP/PS1 mice (see new Figure 8C), making the experiment not possible.

We therefore performed the following experiments: we first tested whether our microfluidic device can be used to assess synapse number. We found that an HTT_SA_ circuit shows more synapses, indicating that circuits-on-a-chip recapitulate in vivo circuit characteristics (new Figure 7A). We then tested APP overexpression as a strategy to increase the quantity of APP within the presynaptic cortical compartment of a WT or HTT_SA_ network (new Figure 7B). WT or HTT_SA_ neurons were transduced with a lentivirus expressing APP at DIV7 and synapse number was measured at DIV12. Overexpressing APP in WT presynaptic cortical neurons decreased synaptic contacts, but overexpressing APP in HTT_SA_ presynaptic cortical neurons restored synaptic contacts back to the levels seen in WT neurons (new Figure 7B). The presynaptic level of APP thus appears to determine synapse number and can be modulated by HTT phosphorylation; this further supports a role for the Akt-HTT-APP pathway in synapse homeostasis.

To circumvent any potential effect of the HTT_SA_ mutation on axonal growth, we transduced WT circuit at DIV8 – when axon growth has ended(Moutaux et al., 2018) – with lentiviruses expressing APP and either an N-terminal HTT construct containing the first 480 amino acids (HTT480-WT) or a construct in which S421 has been mutated into alanine (HTT-480-SA). We found that expressing the HTT-480-SA construct in mature neurons led to an increase in synaptic contacts similar to what we observed in HTT_SA_ neurons differentiated in microchambers (new Figure 7A and C). This suggests that the HTT S421A mutation has no major role in axon growth and/or that the increase of synaptic contacts seen in HTT_SA_ neurons is not due to changes in neurodevelopment but rather on reducing the transport and accumulation of APP at the presynapses. We then investigated the effect of APP overexpression in WT neurons. As in Figure 7B, APP overexpression in WT neurons transduced with HTT-480-WT led to a decrease in the number of synaptic contacts. However, it had no effect in neurons expressing HTT-480-SA, indicating that HTT dephosphorylation reduces the effect of APP overexpression on synapse number (new Figure 7D). We conclude that reducing anterograde axonal transport of APP either during axonal growth or in mature networks is sufficient to modulate synaptic contacts.

Finally, we report that blocking HTT phosphorylation in APPPS1/HTT_SA_ mice leads to a reduction of APP levels in the presynaptic compartment, and a rescue in synapse number and in postsynaptic density length, while having no effect on Ab accumulation or plaques (new Figure 8 and Figure 8—figure supplement 1). Importantly, HTT_SA_ significantly improved spatial and non-spatial memory in APPPS1 mice (new Figure 9 and Figure 9—figure supplement 1).

In addition, it may be important to define the mechanism of APP transport in the axons by further validating previous findings (Colin et al., 2008) such as whether HTT S421 phosphorylation is required for Kinesin-1 recruitment to microtubules and vesicles using HTTS421A/S421A knockin mouse.

We provide subcellular fractionation of cytosolic and vesicular fractions from WT and HTT_SA_ mouse brains showing a trend towards a decreased association of kinesin-1 and HTT_SA_ vesicles (new Figure 3—figure supplement 1). This further supports our previous findings in overexpression experiments.

Furthermore, I am not sure why the authors focused on APP while it is well known that HTT S421 phosphorylation plays a role in transporting vesicles in general, such as BDNF-containing vesicles in addition to APP containing vesicles (Colin et al., 2008).

This is true, but the role of HTT on BDNF transport has been pretty well investigated (Saudou and Humbert, 2016). To quote reviewer 2, “APP trafficking is tied to both basic and neurodegenerative disease processes, and tying these proteins together mechanistically would be a significant advance.”

References:

Gauthier LR, Charrin BC, Borrell-Pages M, Dompierre JP, Rangone H, Cordelieres FP, De Mey J, MacDonald ME, Lessmann V, Humbert S, Saudou F (2004) Huntingtin controls neurotrophic support and survival of neurons by enhancing BDNF vesicular transport along microtubules. Cell 118: 127-38

Lehigh KM, West KM, Ginty DD (2017) Retrogradely Transported TrkA Endosomes Signal Locally within Dendrites to Maintain Sympathetic Neuron Synapses. Cell Rep 19: 86-100

Zala D, Colin E, Rangone H, Liot G, Humbert S, Saudou F (2008) Phosphorylation of mutant huntingtin at S421 restores anterograde and retrograde transport in neurons. Hum Mol Genet 17: 3837-3846